# Invariant neural subspaces maintained by feedback modulation

**Laura B Naumann[1,2]\*, Joram Keijser[1], Henning Sprekeler[1,2]**

[1]Modelling of Cognitive Processes, Technical University of Berlin, Berlin, Germany; [2]Bernstein Center for Computational Neuroscience, Berlin, Germany

**Abstract** Sensory systems reliably process incoming stimuli in spite of changes in context. Most recent models accredit this context invariance to an extraction of increasingly complex sensory features in hierarchical feedforward networks. Here, we study how context-invariant representations can be established by feedback rather than feedforward processing. We show that feedforward neural networks modulated by feedback can dynamically generate invariant sensory representations. The required feedback can be implemented as a slow and spatially diffuse gain modulation. The invariance is not present on the level of individual neurons, but emerges only on the population level. Mechanistically, the feedback modulation dynamically reorients the manifold of neural activity and thereby maintains an invariant neural subspace in spite of contextual variations. Our results highlight the importance of population-level analyses for understanding the role of feedback in flexible sensory processing.

## Editor's evaluation

One of the key questions in sensory neuroscience is how cortical networks extract invariant percepts from variable sensory inputs. While much of the literature focuses on the role of feedforward hierarchical processing for extracting invariant percepts, this study proposes a novel implementation based on top-down feedback. The article analyses the underlying mechanism based on an invariant subspace and presents instantiations of this mechanism at different levels of biophysical realism.

**\*For correspondence:**
laura-bella.naumann@bccn-berlin.de

**Competing interest:** The authors declare that no competing interests exist.

## Introduction

In natural environments, our senses are exposed to a colourful mix of sensory impressions. Behaviourally relevant stimuli can appear in varying contexts, such as variations in lighting, acoustics, stimulus position, or the presence of other stimuli. Different contexts may require different responses to the same stimulus, for example, when the behavioural task changes (context dependence). Alternatively, the same response may be required for different stimuli, for example, when the sensory context changes (context invariance). Recent advances have elucidated how context-*dependent* processing can be performed by recurrent feedback in neural circuits (*Mante et al., 2013*; *Wang et al., 2018a*; *Dubreuil et al., 2020*). In contrast, the role of feedback mechanisms in context-*invariant* processing is not well understood.

In the classical view, stimuli are hierarchically processed towards a behaviourally relevant percept that is invariant to contextual variations. This is achieved by extracting increasingly complex features in a feedforward network (*Kriegeskorte, 2015*; *Zhuang et al., 2021*; *Yamins and DiCarlo, 2016*). Models of such feedforward networks have been remarkably successful at learning complex perceptual tasks (*LeCun et al., 2015*), and they account for various features of cortical sensory representations (*DiCarlo and Cox, 2007*; *Kriegeskorte et al., 2008*; *DiCarlo et al., 2012*; *Hong et al., 2016*; *Cichy et al., 2016*). Yet, these models neglect feedback pathways, which are abundant in sensory

cortex (*Felleman and Van Essen, 1991*; *Markov et al., 2014*) and shape sensory processing in critical ways (*Gilbert and Li, 2013*). Incorporating these feedback loops into models of sensory processing increases their flexibility and robustness (*Spoerer et al., 2017*; *Alamia et al., 2021*; *Nayebi et al., 2021*) and improves their fit to neural data (*Kar et al., 2019*; *Kietzmann et al., 2019*; *Nayebi et al., 2021*). At the neuronal level, feedback is thought to modulate rather than drive local responses (*Sherman and Guillery, 1998*), for instance, depending on behavioural context (*Niell and Stryker, 2010*; *Vinck et al., 2015*; *Kuchibhotla et al., 2017*; *Dipoppa et al., 2018*).

Here, we investigate the hypothesis that feedback modulation provides a neural mechanism for context-invariant perception. To this end, we trained a feedback-modulated network model to perform a context-invariant perceptual task and studied the resulting neural mechanisms. We show that the feedback modulation does not need to be temporally or spatially precise and can be realised by feedback-driven gain modulation in rate-based networks of excitatory and inhibitory neurons. To solve the task, the feedback loop dynamically maintains an invariant subspace in the population representation (*Hong et al., 2016*). This invariance is not present at the single-neuron level. Finally, we find that the feedback conveys a nonlinear representation of the context itself, which can be hard to discern by linear decoding methods.

These findings corroborate that feedback-driven gain modulation of feedforward networks enables context-invariant sensory processing. The underlying mechanism links single-neuron modulation with its function at the population level, highlighting the importance of population-level analyses.

## Results

As a simple instance of a context-invariant task, we considered a dynamic version of the blind source separation problem. The task is to recover unknown sensory sources, such as voices at a cocktail party (*McDermott, 2009*), from sensory stimuli that are an unknown mixture of the sources. In contrast to the classical blind source separation problem, the mixture can change in time, for example, when the speakers move around, thus providing a time-varying sensory context. Because the task requires a dynamic inference of the context, it cannot be solved by feedforward networks (*Figure 1—figure supplement 1*) or standard blind source separation algorithms (e.g. independent component analysis; *Bell and Sejnowski, 1995*; *Hyvärinen and Oja, 2000*). We hypothesised that this dynamic task can be solved by a feedforward network that is subject to modulation from a feedback signal. In our model, the feedback signal is provided by a modulatory system that receives both sensory stimuli and network output (*Figure 1a*).

### Dynamic blind source separation by modulation of feedforward weights

Before we gradually take this to the neural level, we illustrate the proposed mechanism in a simple example, in which the modulatory system provides a time-varying multiplicative modulation of a linear two-layer network (see 'Materials and methods'). For illustration, we used compositions of sines with different frequencies as source signals ($s$, *Figure 1b*, top). These sources were linearly mixed to generate the sensory stimuli ($x$) that the network received as input; $x = A_t s$ (*Figure 1a and b*). The linear mixture ($A_t$) changed over time, akin to varying the location of sound sources in a room (*Figure 1a*). These locations provided a time-varying sensory context that changed on a slower timescale than the sources themselves. The feedforward network had to recover the sources from the mixed sensory stimuli. To achieve this, we trained the modulator to dynamically adjust the weights of the feedforward network ($W_0$) such that the network output ($y$) matches the sources:

$$y = W_t x = (M_t \odot W_0) x$$

$$M_t = \text{modulator}(\text{history of } x, y) .$$

Because the modulation requires a dynamic inference of the context, the modulator is a recurrent neural network. The modulator was trained using supervised learning. Afterwards, its weights were fixed and it no longer had access to the target sources (see 'Materials and methods,' Figure 8). The modulator therefore had to use its recurrent dynamics to determine the appropriate modulatory feedback for the time-varying context, based on the sensory stimuli and the network output. Put

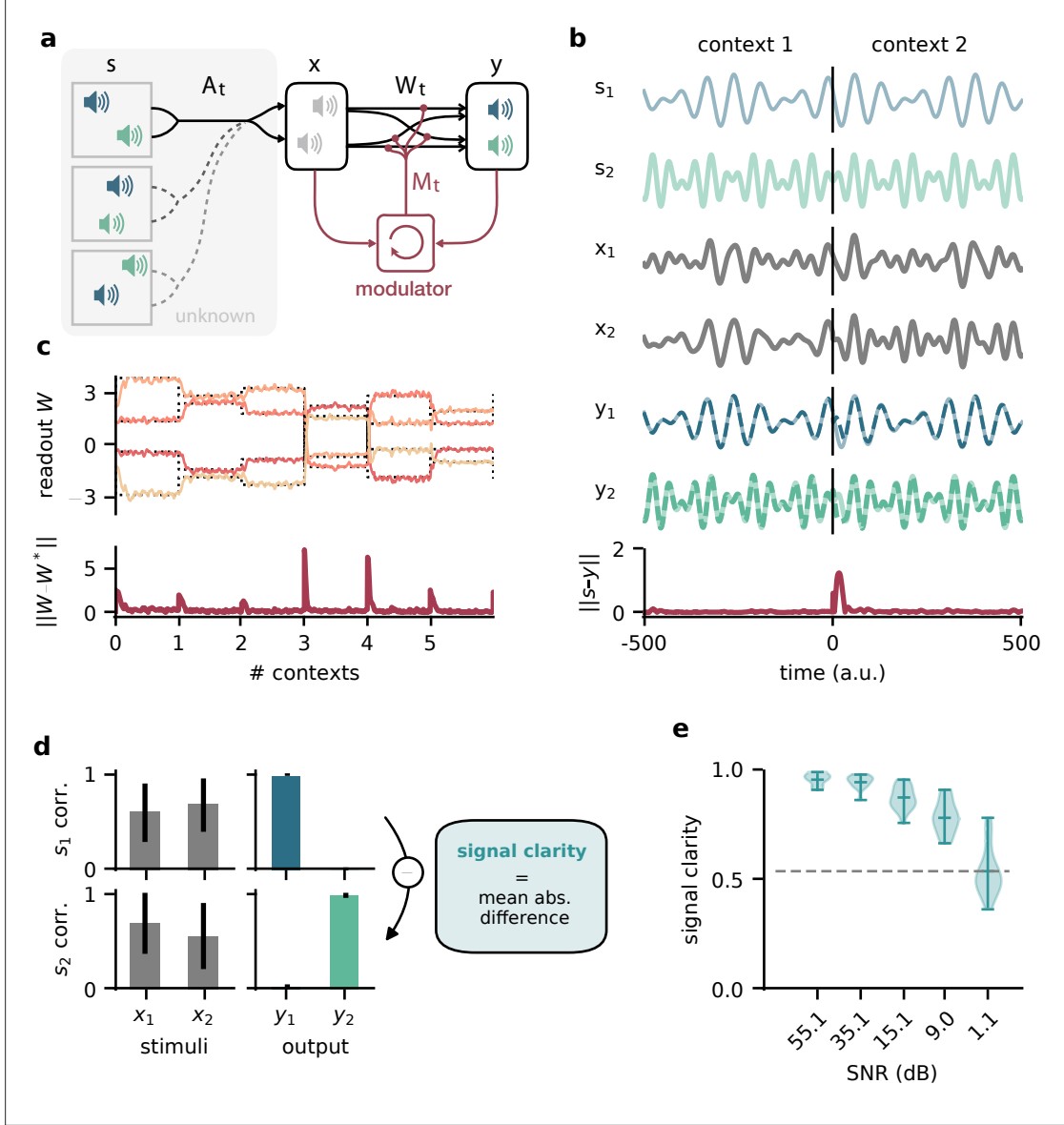

**Figure 1.** Dynamic blind source separation by modulation of feedforward connections. (**a**) Schematic of the feedforward network model receiving feedback modulation from a modulator (a recurrent network). (**b**) Top: sources ($s_{1,2}$), sensory stimuli ($x_{1,2}$), and network output ($y_{1,2}$) for two different source locations (contexts). Bottom: deviation of output from the sources. (**c**) Top: modulated readout weights across six contexts (source locations); dotted lines indicate the true weights of the inverted mixing matrix. Bottom: deviation of readout from target weights. (**d**) Correlation between the sources and the sensory stimuli (left), the network outputs (centre), and calculation of the *signal clarity* (right). Error bars indicate standard deviation across 20 contexts. (**e**) Violin plot of the signal clarity for different noise levels in the sensory stimuli across 20 different contexts.

The online version of this article includes the following figure supplement(s) for figure 1:

**Figure supplement 1.** The dynamic blind source separation task cannot be solved with a feedforward network unless the network receives a sequence of inputs at once. This would require an additional mechanism to retain information over time.

**Figure supplement 2.** Robustness of the feedback-driven modulation mechanism.

**Figure supplement 3.** Model performance for two different sets of source signals.

**Figure supplement 4.** Model performance for three source signals.

**Figure supplement 5.** The modulated network model generalises across frequencies.

**Figure supplement 6.** The modulator learns a model of the sources and contexts, and infers the current context from the stimuli. Testing the network on sources and contexts with different statistics than during training thus impairs its performance.

differently, the modulator had to learn an internal model of the sensory data and the contexts, and use it to establish the desired context invariance in the output.

After learning, the modulated network disentangled the sources, even when the context changed (*Figure 1b*, *Figure 1—figure supplement 1a and b*). Context changes produced a transient error in the network's output, but it quickly resumed matching the sources (*Figure 1b*, bottom). The transient errors occur because the modulator needs time to infer the new context from the time-varying inputs before it can provide the appropriate feedback signal to the feedforward network (*Figure 1—figure supplement 6a*, compare with *Figure 1—figure supplement 1g–i*). The modulated feedforward weights inverted the linear mixture of sources by switching on the same timescale (*Figure 1c*).

To quantify how well the sources were separated, we measured the correlation coefficient of the outputs with each source over several contexts. Consistent with a clean separation, we found that each of the two outputs strongly correlated with only one of the sources. In contrast, the sensory stimuli showed a positive average correlation for both sources as expected given the positive linear mixture (*Figure 1d*, left). We determined the *signal clarity* as the absolute difference between the correlation with the first compared to the second source, averaged over the two outputs, normalised by the sum of the correlations (*Figure 1d*, right; see 'Materials and methods'). The signal clarity thus determines the degree of signal separation, where a value close to 1 indicates a clean separation as in *Figure 1d*. Note that the signal clarity of the sensory stimuli is around 0.5 and can be used as a reference.

We next probed the network's robustness by adding noise to the sensory stimuli. We found that the signal clarity gradually decreased with increasing noise levels, but only degraded to chance performance when the signal-to-noise ratio was close to 1 (1.1 dB, *Figure 1e*, *Figure 1—figure supplement 2e*). The network performance did not depend on the specific source signals (*Figure 1—figure supplement 3*) or the number of sources (*Figure 1—figure supplement 4*) as long as it had seen them during training. Yet, because the network had to learn an internal model of the task, we expected a limited degree of generalisation to new situations. Indeed, the network was able to interpolate between source frequencies seen during training (*Figure 1—figure supplement 5*), but failed on sources and contexts that were qualitatively different (*Figure 1—figure supplement 6b–d*). The specific computations performed by the modulator are therefore idiosyncratic to the problem at hand. Hence, we did not investigate the internal dynamics of the modulator in detail, but concentrated on its effect on the feedforward network.

Since feedback-driven modulation enables flexible context-invariant processing in a simple abstract model, we wondered how this mechanism might be implemented at the neural level. For example, how does feedback-driven modulation function when feedback signals are slow and imprecise? And how does the modulation affect population activity? In the following, we will gradually increase the model complexity to account for biological constraints and pinpoint the population-level mechanisms of feedback-mediated invariance.

## Invariance can be established by slow feedback modulation

Among the many modulatory mechanisms, even the faster ones are believed to operate on timescales of hundreds of milliseconds (*Bang et al., 2020*; *Molyneaux and Hasselmo, 2002*), raising the question if feedback-driven modulation is sufficiently fast to compensate for dynamic changes in environmental context.

To investigate how the timescale of modulation affects the performance in the dynamic blind source separation task, we trained network models, in which the modulatory feedback had an intrinsic timescale that forced it to be slow. We found that the signal clarity degraded only when this timescale was on the same order of magnitude as the timescale of contextual changes (*Figure 2a*). Note that timescales in this model are relative and could be arbitrarily rescaled. While slower feedback modulation produced a larger initial error (*Figure 2b and c*), it also reduced the fluctuations in the readout weights such that they more closely follow the optimal weights (*Figure 2b*). This speed-accuracy trade-off explains the lower and more variable signal clarity for slow modulation (*Figure 2a*) because the signal clarity was measured over the whole duration of a context and the transient onset error dominated over the reduced fluctuations.

To determine architectural constraints on the modulatory system, we asked how these results depended on the input it received. So far, the modulatory system received the feedforward network's inputs (the sensory stimuli) and its outputs (the inferred sources, see *Figure 1a*), but are both of

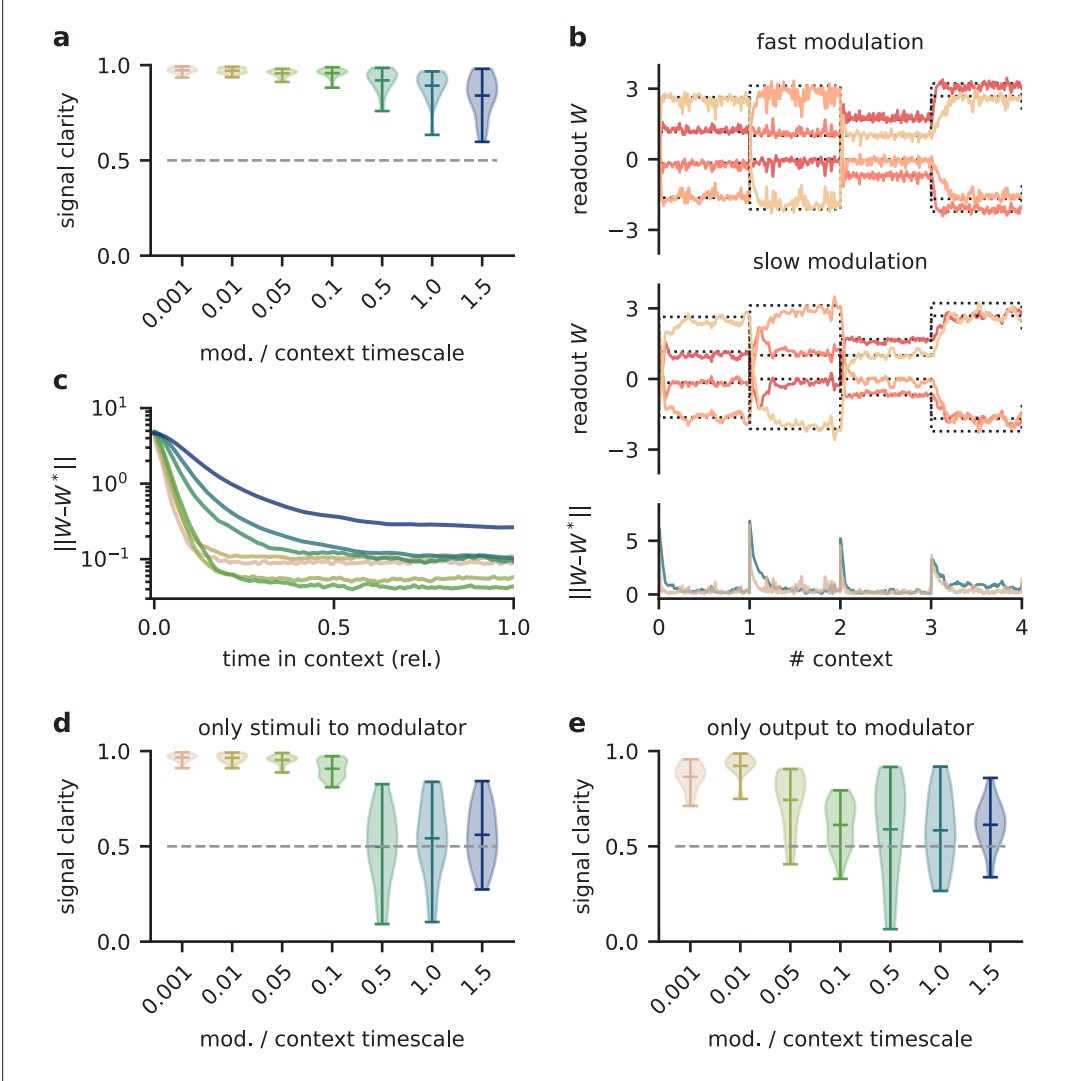

**Figure 2.** The network model is not sensitive to slow feedback modulation. (**a**) Signal clarity in the network output for varying timescales of modulation relative to the intervals at which the source locations change. (**b**) Modulated readout weights across four source locations (contexts) for fast (top) and slow (centre) feedback modulation; dotted lines indicate the optimal weights (the inverse of the mixing matrix). Bottom: deviation of the readout weights from the optimal weights for fast and slow modulation. Colours correspond to the relative timescales in (**a**). Fast and slow timescales are 0.001 and 1, respectively. (**c**) Mean deviation of readout from optimal weights within contexts; averaged over 20 contexts. Colours code for timescale of modulation (see (**a**)). (**d**, **e**) Same as (**a**) but for models in which the modulatory system only received the sensory stimuli $x$ or the network output $y$, respectively.

The online version of this article includes the following figure supplement(s) for figure 2:

**Figure supplement 1.** Robustness to slow feedback modulation depends on the inputs to the modulatory system.

these necessary to solve the task? We found that when the modulatory system only received the sensory stimuli, the model could still learn the task, though it was more sensitive to slow modulation (*Figure 2d*, *Figure 2—figure supplement 1*). When the modulatory system had to rely on the network output alone, task performance was impaired even for fast modulation (*Figure 2e*, *Figure 2—figure supplement 1*). Thus, while the modulatory system is more robust to slow modulation when it receives the network output, the output is not sufficient to solve the task.

Taken together, these results show that the biological timescale of modulatory mechanisms does not pose a problem for flexible feedback-driven processing as long as the feedback modulation changes on a faster timescale than variations in the context. In fact, slow modulation can increase

processing accuracy by averaging out fluctuations in the feedback signal. Nevertheless, slow modulation likely requires the modulatory system to receive both input and output of the sensory system it modulates.

## Invariance can be established by spatially diffuse feedback modulation

Neuromodulators are classically believed to diffusely affect large areas of the brain. Furthermore, signals in the brain are processed by populations of neurons. We wondered if the proposed modulation mechanism is consistent with such biological constraints. We therefore extended the network model such that the sensory stimuli are projected to a population of 100 neurons. A fixed linear

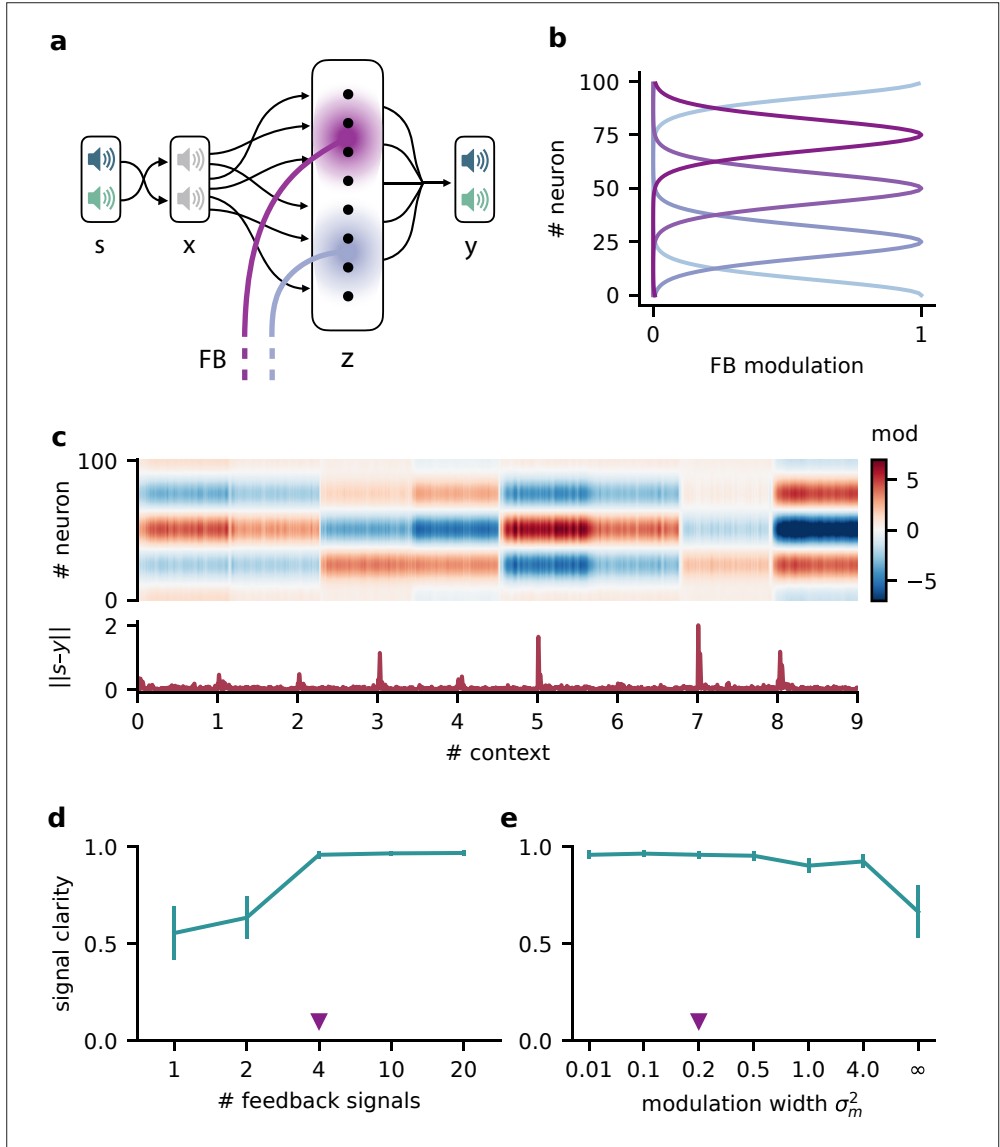

**Figure 3.** Feedback modulation in the model can be spatially diffuse. (**a**) Schematic of the feedforward network with a population that receives diffuse feedback-driven modulation. (**b**) Spatial spread of the modulation mediated by four modulatory feedback signals with a width of 0.2. (**c**) Top: per neuron modulation during eight different contexts. Bottom: corresponding deviation of the network output from sources. (**d**) Mean signal clarity across 20 contexts for different numbers of feedback signals; modulation width is 0.2. Error bars indicate standard deviation. Purple triangle indicates default parameters used in (**c**). (**e**) Same as (**d**) but for different modulation widths; number of feedback signals is 4. The modulation width '∞' corresponds to uniform modulation across the population.

The online version of this article includes the following figure supplement(s) for figure 3:

**Figure supplement 1.** Robustness to the spatial scale of feedback modulation.

readout of this population determined the network output. The neurons in the population received spatially diffuse modulatory feedback (*Figure 3a*) such that the feedback modulation affected neighbouring neurons similarly. We here assume that all synaptic weights to a neuron receive the same modulation, such that the feedback performs a gain modulation of neural activity (*Ferguson and Cardin, 2020*). The spatial specificity of the modulation was determined by the number of distinct feedback signals and their spatial spread (*Figure 3b*, *Figure 3—figure supplement 1a*).

This population-based model with less specific feedback modulation could still solve the dynamic blind source separation task. The diffuse feedback modulation switched when the context changed, but was roughly constant within contexts (*Figure 3c*), as in the simple model. The effective weight from the stimuli to the network output also inverted the linear mixture of the sources (*Figure 3—figure supplement 1d*, compare with *Figure 1c*).

We found that only a few distinct feedback signals were needed for a clean separation of the sources across contexts (*Figure 3d*). Moreover, the feedback could have a spatially broad effect on the modulated population without degrading the signal clarity (*Figure 3e*, *Figure 3—figure supplement 1*), consistent with the low dimensionality of the context.

We conclude that, in our model, neuromodulation does not need to be spatially precise to enable flexible processing. Given that the suggested feedback-driven modulation mechanism works for slow and diffuse feedback signals, it could in principle be realised by neuromodulatory pathways present in the brain.

## Invariance emerges at the population level

Having established that slow and spatially diffuse feedback modulation enables context-invariant processing, we next investigated the underlying mechanisms at the single-neuron and population level. Given that the readout of the population activity was fixed, it is not clear how the

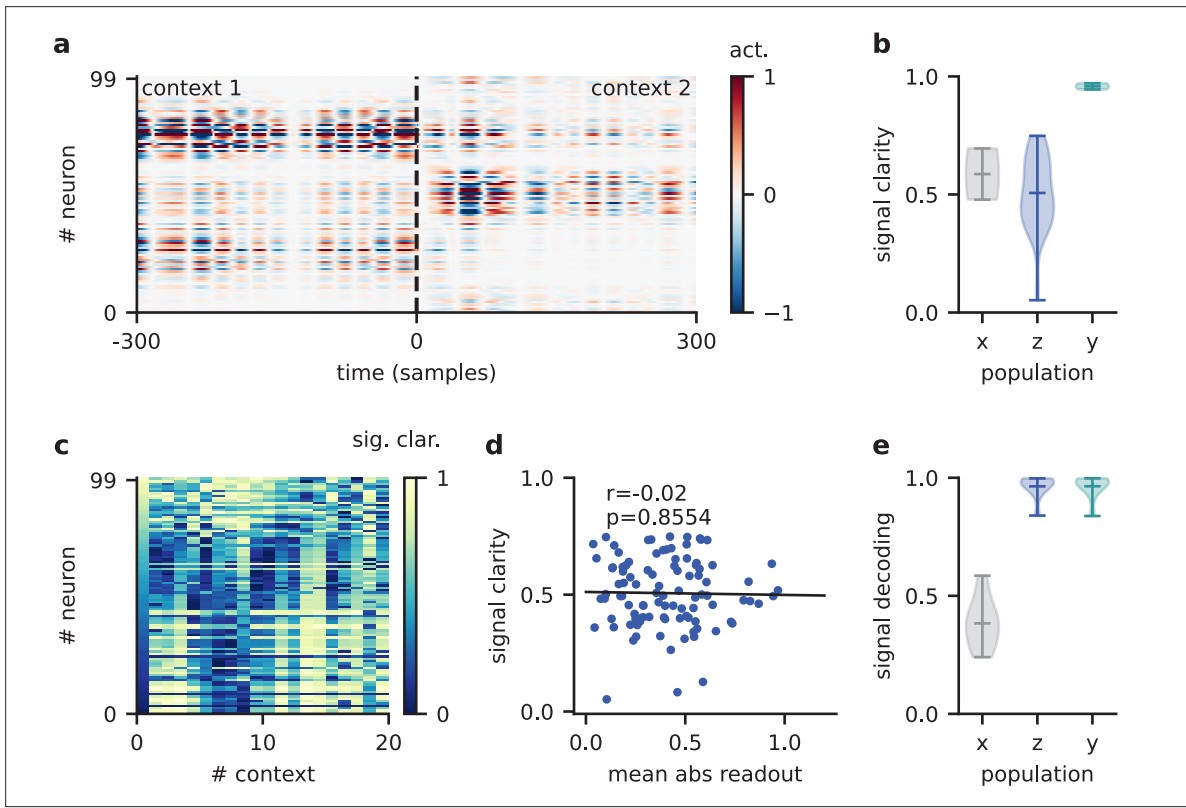

**Figure 4.** Invariance emerges at the population level. (**a**) Population activity in two contexts. (**b**) Violin plot of the signal clarity in the sensory stimuli ($x$), neural population ($z$), and network output ($y$), computed across 20 different contexts. (**c**) Signal clarity of single neurons in the modulated population for different contexts. (**d**) Correlation between average signal clarity over contexts and magnitude of neurons' readout weight. Corresponding Pearson $r$ and $p$-value are indicated in the panel. (**e**) Violin plot of the linear decoding performance of the sources from different stages of the feedforward network, computed across 20 contexts. The decoder was trained on a different set of 20 contexts.

context-dependent modulation of single neurons could give rise to a context-independent network output. One possible explanation is that some of the neurons are context-invariant and are exploited by the readout. However, a first inspection of neural activity indicated that single neurons are strongly modulated by context (*Figure 4a*). To quantify this, we determined the signal clarity for each neuron at each stage of the feedforward network, averaged across contexts (*Figure 4b*). As expected, the signal clarity was low for the sensory stimuli. Intriguingly, the same was true for all neurons of the modulated neural population, indicating no clean separation of the sources at the level of single neurons. Although most neurons had a high signal clarity in some of the contexts, there was no group of neurons that consistently represented one or the other source (*Figure 4c*). Furthermore, the average signal clarity of the neurons did not correlate with their contribution to the readout (*Figure 4d*). Since single-neuron responses were not invariant, context invariance must arise at the population level.

To confirm this, we asked how well the sources could be decoded at different stages of the feedforward network. We trained a single linear decoder of the sources on one set of contexts and tested its generalisation to novel contexts. We found that the decoding performance was poor for the sensory stimuli (*Figure 4e*), indicating that these did not contain a context-invariant representation. In contrast, the sources could be decoded with high accuracy from the modulated population.

This demonstrates that while individual neurons were not invariant, the population activity contained a context-invariant subspace. In fact, the population had to contain an invariant subspace because the fixed linear readout of the population was able to extract the sources across contexts. However, the linear decoding approach shows that this subspace can be revealed from the population activity itself with only a few contexts and no knowledge of how the neural representation is used downstream. The same approach could therefore be used to reveal context-invariant subspaces in neural data from population recordings. Note that the learned readout and the decoder obtained from population activity are not necessarily identical due to the high dimensionality of the population activity compared to the sources.

## Feedback reorients the population representation

The question remains how exactly the context-invariant subspace is maintained by feedback modulation. In contrast to a pure feedforward model of invariant perception (*Kriegeskorte, 2015*; *Yamins and DiCarlo, 2016*), feedback-mediated invariance requires time to establish after contextual changes. Experimentally, hallmarks of this adaptive process should be visible when comparing the population representations immediately after a change and at a later point in time. Our model allows to cleanly separate the early and late representation by freezing the feedback signals in the initial period after a contextual change (*Figure 5a*), thereby disentangling the effects of feedback and context on population activity.

The simulated experiment consisted of three stages: first, the feedback was intact for a particular context and the network outputs closely tracked the sources. Second, the context was changed but the feedback modulation was frozen at the same value as before. As expected, this produced deviations of the output from the sources. Third, for the same context the feedback modulation was turned back on, which reinstated the source signals in the output. In this experiment, we used pure sines as signals for visualisation purposes (*Figure 5a and c*). To visualise the population activity in the three stages of the experiment, we considered the space of the two readout dimensions and the first principal component (*Figure 5b*). We chose this space rather than, for example, the first three principal components (*Figure 5—figure supplement 1*), because it provides an intuitive illustration of the invariant subspace.

Because the sources were two-dimensional, the population activity followed a pattern within a two-dimensional subspace (*Figure 5b*, left; *Figure 5—figure supplement 1a*). For intact feedback, this population activity matched the sources when projected onto the readout (*Figure 5c*, left). Changing the context while freezing the feedback rotated and stretched this representation within the same subspace, such that the readout did not match the sources (*Figure 5b and c*, centre). Would turning the feedback modulation back on simply reverse this transformation to re-establish an invariant subspace? We found that this was not the case. Instead, the feedback rotated the representation out of the old subspace (*Figure 5b*, right), thereby reorienting it into the invariant readout (*Figure 5c*, right).

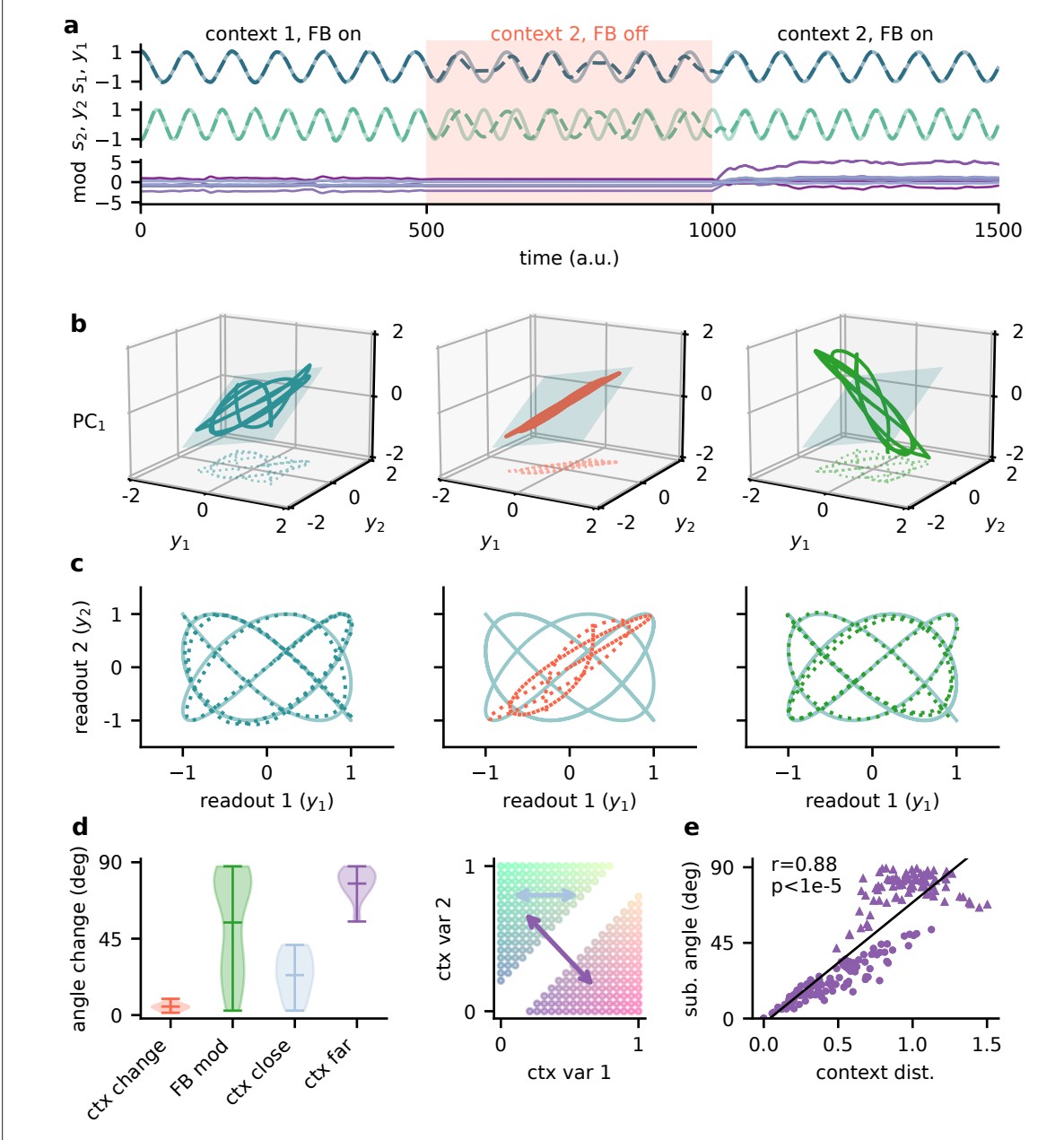

**Figure 5.** Feedback reorients the population representation. (**a**) Network output (top) and feedback modulation (bottom) for two contexts. The feedback modulation is frozen for the initial period after the context changes. (**b**) Population activity in the space of the two readout axes and the first principal component. Projection onto the readout is indicated at the bottom (see (**c**)). The signal representation is shown for different phases of the experiment. Left: context 1 with intact feedback; centre: context 2 with frozen feedback; right: context 2 with intact feedback. The blue plane spans the population activity subspace in context 1 (left). (**c**) Same as (**b**), but projected onto the readout space (dotted lines in (**b**)). The light blue trace corresponds to the sources. (**d**) Left: change in subspace orientation across 40 repetitions of the experiment, measured by the angle between the original subspace and the subspace for context changes (ctx change), feedback modulation (FB mod), and feedback modulation for similar contexts (ctx close) or dissimilar contexts (ctx far). Right: two-dimensional context space, defined by the coefficients in the mixing matrix. Arrows indicate similar (light blue) and dissimilar contexts (purple). (**e**) Distance between pairs of contexts versus the angle between population activity subspaces for these contexts. Circles indicate similar contexts (from the same side of the diagonal, see (**d**)) and triangles dissimilar contexts (from different sides of the diagonal). Pearson $r$ and $p$-value indicated in the panel.

The online version of this article includes the following figure supplement(s) for figure 5:

**Figure supplement 1.** Principal component (PC) analysis captures the low-dimensional population subspaces and the subspace reorientation with feedback.

To quantify the transformation of the population representation, we repeated this experiment multiple times and determined the angle between the neural subspaces. Consistent with the illustration in *Figure 5b*, changing the context did not change the subspace orientation, whereas unfreezing the feedback caused a consistent reorientation (*Figure 5d*). The magnitude of this subspace reorientation depended on the similarity of the old and new context. Similar contexts generally evoked population activity with similar subspace orientations (*Figure 5d and e*). This highlights that there is a consistent mapping between contexts and the resulting low-dimensional population activity.

In summary, the role of feedback-driven modulation in our model is to reorient the population representation in response to changing contexts such that an invariant subspace is preserved.

## The mechanism generalises to a hierarchical Dalean network

So far, we considered a linear network, in which neural activity could be positive and negative. Moreover, feedback modulation could switch the sign of the neurons' downstream influence, which is inconsistent with Dale's principle. We wondered if the same population-level mechanisms would operate in a Dalean network, in which feedback is implemented as a positive gain modulation. Although gain modulation is a broadly observed phenomenon that is attributed to a range of cellular mechanisms (*Ferguson and Cardin, 2020*; *Salinas and Thier, 2000*), its effect at the population level is less clear (*Shine et al., 2021*).

We extended the feedforward model as follows (*Figure 6a*): first, all neurons had positive firing rates. Second, we split the neural population ($z$ in the previous model) into a 'lower-level' ($z^{\mathrm{L}}$) and 'higher-level' population ($z^{\mathrm{H}}$). The lower-level population served as a neural representation of the sensory stimuli, whereas the higher-level population was modulated by feedback. This allowed a direct comparison between a modulated and an unmodulated neural population. It also allowed us to include Dalean weights between the two populations. Direct projections from the lower-level to the higher-level population were excitatory. In addition, a small population of local inhibitory neurons provided feedforward inhibition to the higher-level population. Third, the modulation of the higher-level population was implemented as a local gain modulation that scaled the neural responses. As a specific realisation of gain modulation, we assumed that feedback targeted inhibitory interneurons (e.g. in layer 1; *Abs et al., 2018*; *Ferguson and Cardin, 2020*; *Cohen-Kashi Malina et al., 2021*) that mediate the modulation in the higher-level population (e.g. via presynaptic inhibition; *Pardi et al., 2020*; *Naumann and Sprekeler, 2020*). This means that stronger feedback decreased the gain of neurons (*Figure 4b*). We will refer to these modulatory interneurons as modulation units $m$ (green units in *Figure 4a*).

We found that this biologically more constrained model could still learn the context-invariant processing task (*Figure 6—figure supplement 1a and b*). Notably, the network's performance did not depend on specifics of the model architecture, such as the target of the modulation or the number of inhibitory neurons (*Figure 6—figure supplement 1c–e*). In analogy to the previous model, the gain modulation of individual neurons changed with the context and thus enabled the flexible processing required to account for varying context (*Figure 4c*). The average gain over contexts was similar across neurons, whereas within a context the gains were broadly distributed (*Figure 4d*).

To verify if the task is solved by the same population-level mechanism, we repeated our previous analyses on the single-neuron and population level. Indeed, all results generalised to the Dalean network with feedback-driven gain modulation (compare with *Figures 4–6*). Single neurons in the higher- and lower-level population were not context-invariant (*Figure 6e*), but the higher-level population contained a context-invariant subspace (*Figure 6f*). This was not the case for the lower-level population, underscoring that invariant representations do not just arise from projecting the sensory stimuli into a higher dimensional space. Instead, the invariant subspace in the higher-level population was again maintained by the feedback modulation, which reoriented the population activity in response to context changes (*Figure 6g*).

## Feedback conveys a nonlinear representation of the context

Since single neurons in the higher-level population were not invariant to context, the population representation must also contain contextual information. Indeed, contextual variables could be linearly decoded from the higher-level population activity (*Figure 7a*). In contrast, decoding the context from the lower-level population gave much lower accuracy. This shows that the contextual information is

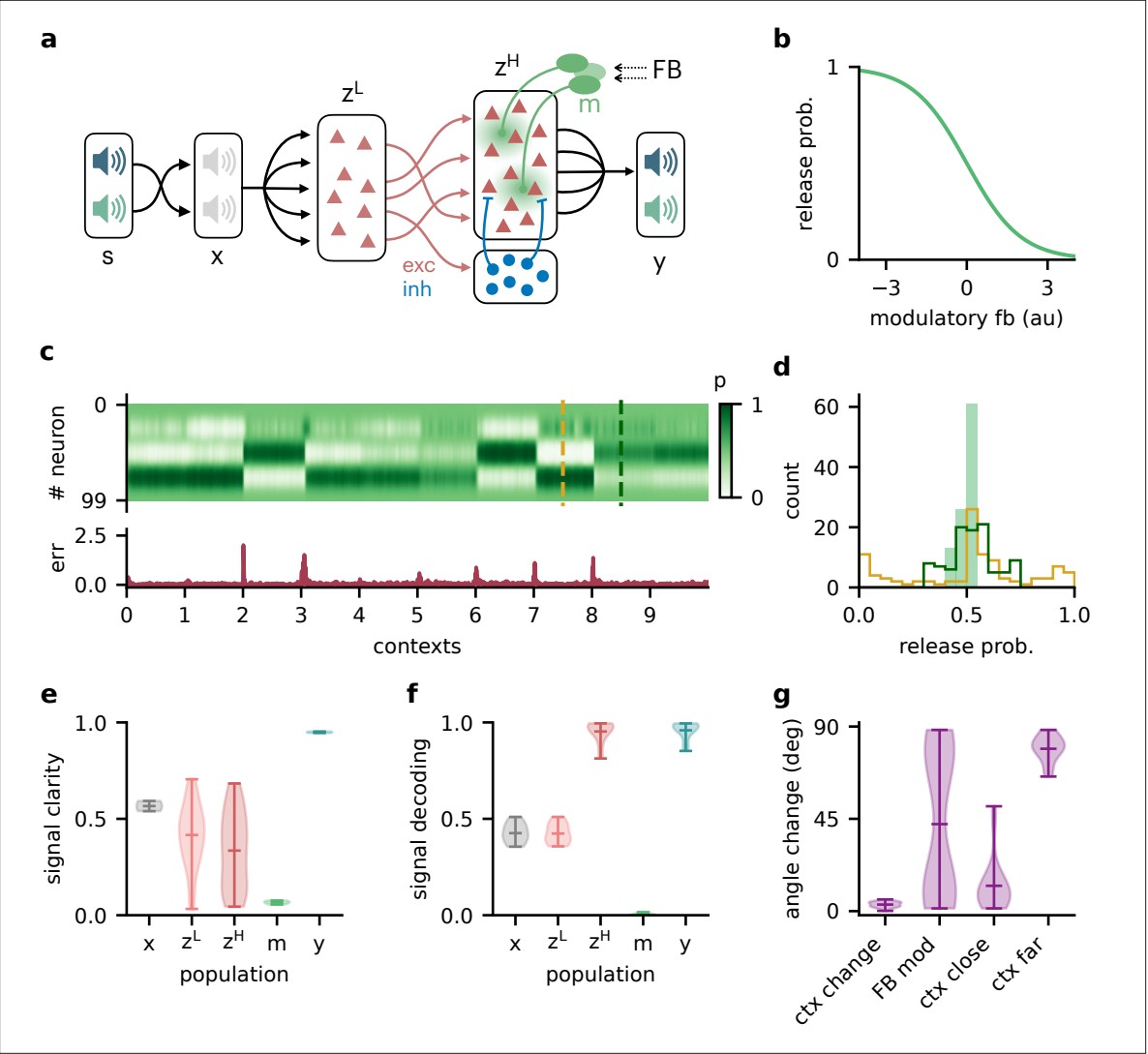

**Figure 6.** Feedback-driven gain modulation in a hierarchical rate network. (**a**) Schematic of the Dalean network comprising a lower- and higher-level population ($z^L$ and $z^H$), a population of local inhibitory neurons (blue), and diffuse gain modulation mediated by modulatory interneurons (green). (**b**) Decrease in gain (i.e. release probability) with stronger modulatory feedback. (**c**) Top: modulation of neurons in the higher-level population for 10 different contexts. Bottom: corresponding deviation of outputs $y$ from sources $s$. (**d**) Histogram of neuron-specific release probabilities averaged across 20 contexts (filled, light green) and during two different contexts (yellow and dark green, see (**c**)). (**e**) Violin plot of signal clarity at different stages of the Dalean model: sensory stimuli ($x$), lower-level ($z^L$) and higher-level population ($z^H$), modulatory units ($m$), and network output ($y$), computed across 20 contexts (compare with *Figure 4a*). (**f**) Violin plot of linear decoding performance of the sources from the same stages as in (**e**) (compare with *Figure 4d*). (**g**) Feedback modulation reorients the population activity (compare with *Figure 5d*).

The online version of this article includes the following figure supplement(s) for figure 6:

**Figure supplement 1.** The Dalean network can learn the dynamic blind source separation task, and the performance does not depend on specifics of the model architecture.

not just inherited from the sensory stimuli but conveyed by the feedback via the modulatory units. We therefore expected that the modulatory units themselves would contain a representation of the context. To our surprise, decoding accuracy on the modulatory units was low. This seems counterintuitive, especially since the modulatory units clearly covaried with the contextual variables (*Figure 7b*). To understand these seemingly conflicting results, we examined how the context was represented in the activity of the modulation units.

We found that the modulation unit activity did encode the contextual variables, albeit in a nonlinear way (*Figure 7c*). The underlying reason is that the feedback modulation needs to remove contextual variations, which requires nonlinear computations. Specifically, the blind source separation task

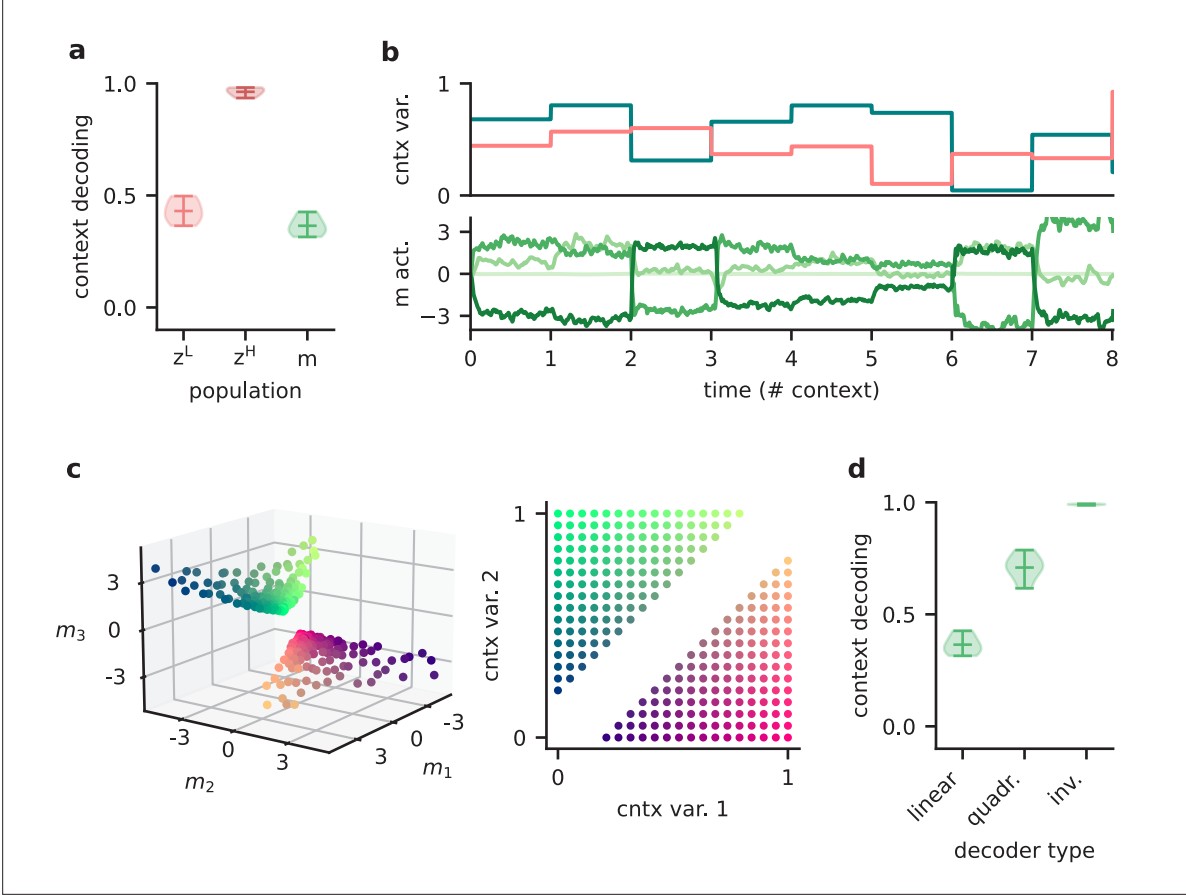

**Figure 7.** Feedback conveys a nonlinear representation of the context. (**a**) Linear decoding performance of the context (i.e. mixing) from the network. (**b**) Context variables (e.g. source locations, top) and activity of modulatory interneurons (bottom) over contexts; one of the modulatory interneurons is silent in all contexts. (**c**) Left: activity of the three active modulatory interneurons (see (**b**)) for different contexts. The context variables are colour-coded as indicated on the right. (**d**) Performance of different decoders trained to predict the context from the modulatory interneuron activity. Decoder types are a linear decoder, a decoder on a quadratic expansion, and a linear decoder trained to predict the inverse of the mixing matrix.

requires an inversion of the linear mixture of sources. Consistent with this idea, nonlinear decoding approaches performed better (*Figure 7d*). In fact, the modulatory units contained a linear representation of the 'inverse context' (i.e. the inverse mixing matrix, see 'Materials and methods').

In summary, the higher-level population provides a linear representation not only of the stimuli, but also of the context. In contrast, the modulatory units contained a nonlinear representation of the context, which could not be extracted by linear decoding approaches. We speculate that if contextual feedback modulation is mediated by interneurons in layer 1, they should represent the context in a nonlinear way.

## Discussion

Accumulating evidence suggests that sensory processing is strongly modulated by top-down feedback projections (*Gilbert and Li, 2013*; *Keller and Mrsic-Flogel, 2018*). Here, we demonstrate that feedback-driven gain modulation of a feedforward network could underlie stable perception in varying contexts. The feedback can be slow, spatially diffuse, and low-dimensional. To elucidate how the context invariance is achieved, we performed single-neuron and population analyses. We found that invariance was not evident at the single-neuron level, but only emerged in a subspace of the population representation. The feedback modulation dynamically transformed the manifold of neural activity patterns such that this subspace was maintained across contexts. Our results provide further support that gain modulation at the single-cell level enables nontrivial computations at the population level (*Failor et al., 2021*; *Shine et al., 2021*).

## Invariance in sensory processing

As an example of context-invariant sensory processing, we chose a dynamic variant of the blind source separation task. This task is commonly illustrated by a mixture of voices at a cocktail party (*Cherry, 1953*; *McDermott, 2009*). For auditory signals, bottom-up mechanisms of frequency segregation can provide a first processing step for the separation of multiple sound sources (*Bronkhorst, 2015*; *McDermott, 2009*). However, separating more complex sounds requires additional active top-down processes (*Parthasarathy et al., 2020*; *Oberfeld and Klöckner-Nowotny, 2016*). In our model, top-down feedback guides the source separation itself, while the selection of a source would occur at a later processing stage – consistent with recent evidence for 'late selection' (*Brodbeck et al., 2020*; *Har-Shai Yahav and Zion Golumbic, 2021*).

Although blind source separation is commonly illustrated with auditory signals, the suggested mechanism of context-invariant perception is not limited to a given sensory modality. The key nature of the task is that it contains stimulus dimensions that need to be encoded (the sources) and dimensions that need to be ignored (the context). In visual object recognition, for example, the identity of visual objects needs to be encoded, while contextual variables such as size, location, orientation, or surround need to be ignored. Neural hallmarks of invariant object recognition are present at the population level (*DiCarlo and Cox, 2007*; *DiCarlo et al., 2012*; *Hong et al., 2016*), and to some extent also on the level of single neurons (*Quiroga et al., 2005*). Classically, the emergence of invariance has been attributed to the extraction of invariant features in feedforward networks (*Riesenhuber and Poggio, 1999*; *Wiskott and Sejnowski, 2002*; *DiCarlo and Cox, 2007*; *Kriegeskorte, 2015*), but recent work also highlights the role of recurrence and feedback (*Gilbert and Li, 2013*; *Kar et al., 2019*; *Kietzmann et al., 2019*; *Thorat et al., 2021*). Here, we focused on the role of feedback, but clearly, feedforward and feedback processes are not mutually exclusive and likely work in concert to create invariance. Their relative contribution to invariant perception requires further studies and may depend on the invariance in question.

Similarly, how invariance can be learned will depend on the underlying mechanism. The feedback-driven mechanism we propose is reminiscent of meta-learning consisting of an inner and an outer loop (*Hochreiter et al., 2001*; *Wang et al., 2018b*). In the inner loop, the modulatory system infers the context to modulate the feedforward network accordingly. This process is unsupervised. In the outer loop, the modulatory system is trained to generalise across contexts. Here, we performed this training using supervised learning, which requires the modulatory system to experience the sources in isolation (or at least obtain an error signal). Such an identification of the individual sources could, for example, be aided by other sensory modalities (*McDermott, 2009*). However, the optimisation of the modulatory system does not necessarily require supervised learning. It could also be guided by task demands via reinforcement learning or by unsupervised priors such as a non-Gaussianity of the outputs.

## Mechanisms of feedback-driven gain modulation

There are different ways in which feedback can affect local processing. Here, we focused on gain modulation (*McAdams and Maunsell, 1999*; *Reynolds and Heeger, 2009*; *Vinck et al., 2015*). Neuronal gains can be modulated by a range of mechanisms (*Ferguson and Cardin, 2020*; *Shine et al., 2021*). In our model, the mechanism needs to satisfy a few key requirements: (i) the modulation is not uniform across the population, (ii) it operates on a timescale similar to that of changes in context, and (iii) it is driven by a brain region that has access to the information needed to infer the context.

Classical neuromodulators such as acetylcholine (*Disney et al., 2007*; *Kawai et al., 2007*), dopamine (*Thurley et al., 2008*), or serotonin (*Azimi et al., 2020*) are signalled through specialised neuromodulatory pathways from subcortical nuclei (*van den Brink et al., 2019*). These neuromodulators can control the neural gain depending on behavioural states such as arousal, attention, or expectation of rewards (*Ferguson and Cardin, 2020*; *Hasselmo and McGaughy, 2004*; *Bayer and Glimcher, 2005*; *Polack et al., 2013*; *Kuchibhotla et al., 2017*). Their effect is typically thought to be brain-wide and long-lasting, but recent advances in measurement techniques (*Sabatini and Tian, 2020*; *Lohani et al., 2020*) indicate that it could be area- or even layer-specific, and vary on sub-second timescales (*Lohani et al., 2020*; *Bang et al., 2020*; *Poorthuis et al., 2013*; *Pinto et al., 2013*).

More specific feedback projections arrive in layer 1 of the cortex, where they target the distal dendrites of pyramidal cells and inhibitory interneurons (*Douglas and Martin, 2004*; *Roth et al.,*

*2016*; *Marques et al., 2018*). Dendritic input can change the gain of the neural transfer function on fast timescales (*Larkum et al., 2004*; *Jarvis et al., 2018*). The spatial scale of the modulation will depend on the spatial spread of the feedback projections and the dendritic arbourisation. Feedback to layer 1 interneurons provides an alternative mechanism of local gain control. In particular, neuron-derived neurotrophic factor-expressing interneurons (NDNF) in layer 1 receive a variety of top-down feedback projections and produce GABAergic volume transmission (*Abs et al., 2018*), thereby down-regulating synaptic transmission (*Miller, 1998*; *Laviv et al., 2010*). This gain modulation can act on a timescale of hundreds of milliseconds (*Branco and Staras, 2009*; *Urban-Ciecko et al., 2015*; *Cohen-Kashi Malina et al., 2021*; *Molyneaux and Hasselmo, 2002*), and, although generally considered diffuse, can also be synapse type-specific (*Chittajallu et al., 2013*).

The question remains where in the brain the feedback signals originate. Our model requires the responsible network to receive feedforward sensory input to infer the context. In addition, feedback inputs from higher-level sensory areas to the modulatory system allow a better control of the modulated network state. Higher-order thalamic nuclei are ideally situated to integrate different sources of sensory inputs and top-down feedback (*Sampathkumar et al., 2021*) and mediate the resulting modulation by targeting layer 1 of lower-level sensory areas (*Purushothaman et al., 2012*; *Roth et al., 2016*; *Sherman, 2016*). In our task setting, the inference of the context requires the integration of sensory signals over time and therefore recurrent neural processing. For this kind of task, thalamus may not be the site of contextual inference because it lacks the required recurrent connectivity (*Halassa and Sherman, 2019*). However, contextual inference may be performed by higher-order cortical areas and could either be relayed back via the thalamus or transmitted directly, for example, via cortico-cortical feedback connections.

## Testable predictions

Our model makes several predictions that could be tested in animals performing invariant sensory perception. Firstly, our model indicates that invariance across contexts may only be evident at the neural population level, but not on the single-cell level. Probing context invariance at different hierarchical stages of sensory processing may therefore require population recordings and corresponding statistical analyses such as neural decoding (*Glaser et al., 2020*). Secondly, we assumed that this context invariance is mediated by feedback modulation. The extent to which context invariance is enabled by feedback on a particular level of the sensory hierarchy could be studied by manipulating feedback connections. Since layer 1 receives a broad range of feedback inputs from different sources, this may require targeted manipulations. If no effect of feedback on context invariance is found, this may either indicate that feedforward mechanisms dominate or that the invariance in question is inherited from an earlier stage, in which it may well be the result of feedback modulation. Given that feedback is more pronounced in higher cortical areas (*McAdams and Maunsell, 1999*; *Pardi et al., 2020*), we expect that the contribution of feedback may play a larger role for the more complex forms of invariance further up in the sensory processing hierarchy. Thirdly, for feedback to mediate context invariance, the feedback projections need to contain a representation of the contextual variables. Our findings suggest, however, that the detection of this representation may require a nonlinear decoding method. Finally, a distinguishing feature of feedback and feedforward mechanisms is that feedback mechanisms take more time. We found that immediately following a sudden contextual change, the neuronal representation initially changes within the manifold associated with the previous context. Later, the feedback reorients the manifold to re-establish the invariance on the population level. Whether these dynamics are a signature of feedback processing or also present in feedforward networks will be an interesting question for future work.

## Comparison to prior work

Computational models have implicated neuronal gain modulation for a variety of functions (*Salinas and Sejnowski, 2001*; *Reynolds and Heeger, 2009*). Even homogeneous changes in neuronal gain can achieve interesting population effects (*Shine et al., 2021*), such as orthogonalisation of sensory responses (*Failor et al., 2021*). More heterogeneous gain modulation provides additional degrees of freedom that enables, for example, attentional modulation (*Reynolds and Heeger, 2009*; *Carandini and Heeger, 2011*), coordinate transformations (*Salinas and Thier, 2000*), and – when amplified by recurrent dynamics – a rich repertoire of neural trajectories (*Stroud et al., 2018*). Gain modulation has

also been suggested as a means to establish invariant processing (*Salinas and Abbott, 1997*), as a biological implementation of dynamic routing (*Olshausen et al., 1993*). While the modulation in these models of invariance can be interpreted as an abstract form of feedback, the resulting effects on the population level were not studied.

An interesting question is by which mechanisms the appropriate gain modulation is computed. In previous work, gain factors were often learned individually for each context, for example, by gradient descent or Hebbian plasticity (*Olshausen et al., 1993*; *Salinas and Abbott, 1997*; *Stroud et al., 2018*), mechanisms that may be too slow to achieve invariance on a perceptual timescale (*van Hemmen and Sejnowski, 2006*). In our model, by contrast, the modulation is dynamically controlled by a recurrent network. Once it has been trained, such a recurrent modulatory system can rapidly infer the current context and provide an appropriate feedback signal on a timescale only limited by the modulatory mechanism.

### Limitations and future work

In our model, we simplified many aspects of sensory processing. Using simplistic sensory stimuli – compositions of sines – allowed us to focus on the mechanisms at the population level, while avoiding the complexities of natural sensory stimuli and deep sensory hierarchies. Although we do not expect conceptual problems in generalising our results to more complex stimuli, such as speech or visual stimuli, the associated computational challenges are substantial. For example, the feedback in our model was provided by a recurrent network, whose parameters were trained by backpropagating errors through the network and through time. This training process can get very challenging for large networks and long temporal dependencies (*Bengio et al., 1994*; *Pascanu et al., 2013*).

In our simulations, we trained the whole model – the modulatory system, the sensory representation, and the readout. For the simplistic stimuli we used, we observed that the training process mostly concentrated on optimising the modulatory system and readout, while a random mapping of sensory stimuli to neural representations seemed largely sufficient to solve the task. For more demanding stimuli, we expect that the sensory representation the modulatory system acts upon may become more important. A well-suited representation could minimise the need for modulatory interventions (*Finn et al., 2017*), in a coordinated interaction of feedforward and feedback.

To understand the effects of feedback modulation on population representations, we included biological constraints in the feedforward network and the structure of the modulatory feedback. However, we did not strive to provide a biologically plausible implementation for the computation of the appropriate feedback signals and instead used an off-the-shelf recurrent neural network (*Hochreiter and Schmidhuber, 1997*). The question how these signals could be computed in a biologically plausible way remains for future studies. The same applies to the question how the appropriate feedback signals can be learned by local learning rules (*Lillicrap et al., 2020*) and how neural representations and modulatory systems learn to act in concert.

## Materials and methods

To study how feedback-driven modulation can enable flexible sensory processing, we built models of feedforward networks that are modulated by feedback. The feedback was dynamically generated by a modulatory system, which we implemented as a recurrent network. The weights of the recurrent network were trained such that the feedback modulation allowed the feedforward network to solve a flexible invariant processing task.

### The dynamic blind source separation task

As an instance of flexible sensory processing, we used a dynamic variant of blind source separation. In classical blind source separation, two or more unknown time-varying sources $\vec{s}(t)$ need to be recovered from a set of observations (i.e. sensory stimuli) $\vec{x}(t)$. The sensory stimuli are composed of an unknown linear mixture of the sources such that $\vec{x}(t) = A\vec{s}(t)$ with a fixed mixing matrix $A$. Recovering the sources requires to find weights $W$ such that $W\vec{x}(t) \approx \vec{s}(t)$. Ideally, $W$ is equal to the pseudo-inverse of the unknown mixing matrix $A$, up to permutations.

**Table 1.** Default parameters of the dynamic blind source separation task.

| Parameter | Symbol | Value |
|---|---|---|
| Number of signals | $n_s$ | 2 |
| Number of samples in context | $n_t$ | 1000 |
| Additive noise | $\sigma_n$ | 0.001 |
| Sampling frequency | $f_s$ | 8 KHz |

In our dynamic blind source separation task, we model variations in the stimulus context by changing the linear mixture over time – albeit on a slower timescale than the time-varying signals. Thus, the sensory stimuli are constructed as

$$\vec{x}(t) = A(t)\vec{s}(t) + \sigma_n \vec{\xi}(t) \quad , \tag{1}$$

where $A(t)$ is a time-dependent mixing matrix and $\sigma_n$ is the amplitude of additive white noise $\vec{\xi}(t)$. The time-dependent mixing matrix determines the current context and was varied in discrete time intervals $n_t$, meaning that the mixing matrix $A(t)$ (i.e. the context) was constant for $n_t$ samples before it changed. The goal of the dynamic blind source separation task is to recover the original signal sources $\vec{s}$ from the sensory stimuli $\vec{x}$ across varying contexts. Thus, the network model output needs to be invariant to the specific context of the sources. Note that while the context was varied, the sources themselves were the same throughout the task, unless stated otherwise. Furthermore, in the majority of experiments the number of source signals and sensory stimuli was $n_s = 2$. A list of default parameters for the dynamic blind source separation task can be found in **Table 1**.

## Source signals
As default source signals, we used two compositions of two sines each ('chords') with a sampling rate of $f_s = 8000$ Hz that can be written as

$$s_1(t) = \sin(2\pi f_{11} t/f_s) + \sin(2\pi f_{12} t/f_s) \tag{2}$$

$$s_2(t) = \sin(2\pi f_{21} t/f_s) + \sin(2\pi f_{22} t/f_s) \tag{3}$$

with frequencies $f_{11} = 100$ Hz, $f_{12} = 125$ Hz, $f_{21} = 150$ Hz, and $f_{22} = 210$ Hz. Note that in our model we measure time as the number of samples from the source signals, meaning that timescales are relative and could be arbitrarily rescaled.

In **Figure 5**, we used pure sine signals with frequency $f$ for visualisation purposes: $s_i = \sin(2\pi ft/f_s)$. We also validated the model on signals that are not made of sine waves, as a sawtooth and a square wave signal (**Figure 1—figure supplement 4**). Unless stated otherwise, the same signals were used for training and testing the model.

## Time-varying contexts
We generated the mixing matrix $A$ for each context by drawing random weights from a uniform distribution between 0 and 1, allowing only positive mixtures of the sources. Unless specified otherwise, we sampled new contexts for each training batch and for the test data, such that the training and test data followed the same distribution without necessarily being the same. The dimension of the mixing matrices was determined by number of signals $n_s$ such that $A$ was of shape $n_s \times n_s$. To keep the overall amplitude of the sensory stimuli in a similar range across different mixtures, we normalised the row sums of each mixing matrix to one. In the case of $n_s = 2$, this implies that the contexts (i.e. the mixing matrices) are drawn from a two-dimensional manifold (see **Figure 8**, bottom left). In addition, we only used the randomly generated mixing matrices whose determinant was larger than some threshold value. We did this to ensure that each signal mixture was invertible and that the weights needed to invert the mixing matrix were not too extreme. A threshold value of 0.2 was chosen based on visual inspection of the weights from the inverted mixing matrix.

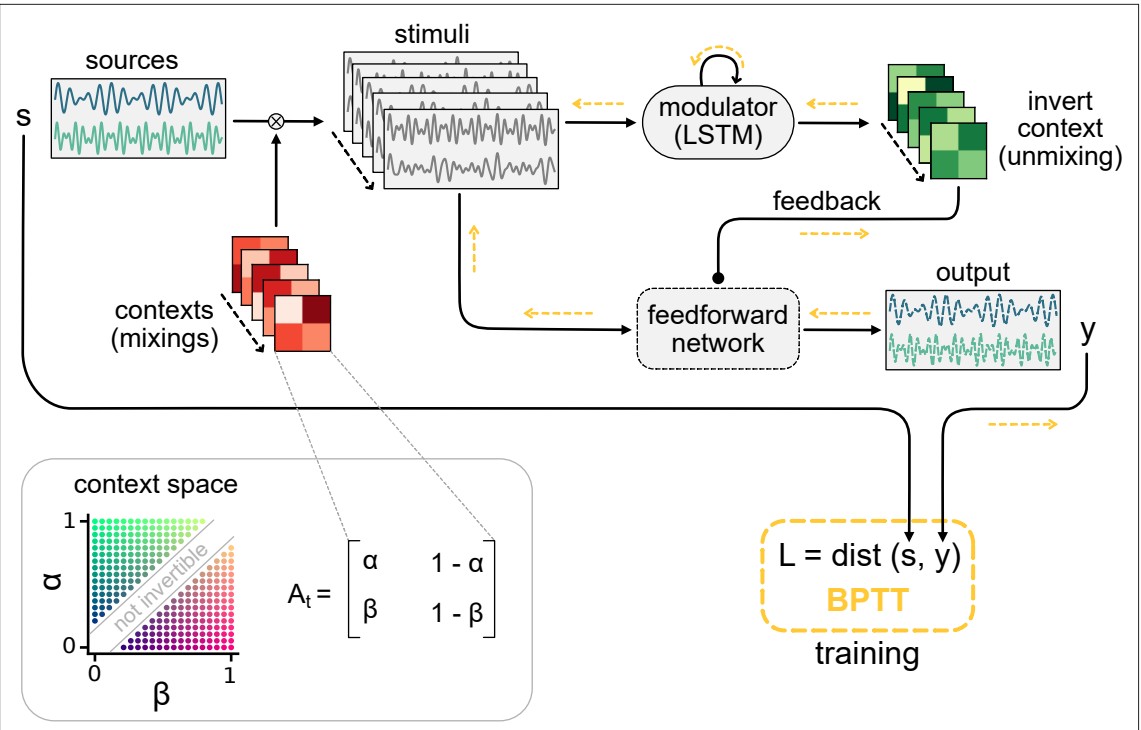

**Figure 8.** Schematic of the dynamic blind source separation task, context space, and the modulated feedforward network. Information flow is indicated by black arrows, and the flow of the error during training with backpropagation through time (BPTT) is shown in yellow.

**Table 2.** Default parameters of the network models.

| Parameter | Symbol | Value |
| --- | --- | --- |
| Number of hidden units in long-short-term memory network | $N_h$ | 100 |
| Number of units in middle layer z | $N_z$ | 100 |
| Number of distinct feedback signals | $N_{FB}$ | 4 |
| Number of neurons in lower-level population | $N_L$ | 40 |
| Number of neurons in higher-level population | $N_H$ | 100 |
| Number of inhibitory neurons | $N_I$ | 20 |
| Timescale of modulation | $\tau$ | 100 |
| Spatial spread of modulation | $\sigma_m^2$ | 0.2 |

## Modulated feedforward network models

Throughout this work, we modelled feedforward networks of increasing complexity. Common to all networks was that they received the sensory stimuli $\vec{x}$ and should provide an output $\vec{y}$ that matches the source signals $\vec{s}$. In the following, we first introduce the simplest model variant and how it is affected by feedback from the modulatory system, and subsequently describe the different model extensions.

### Modulation of feedforward weights by a recurrent network

In the simplest feedforward network, the network output $\vec{y}(t)$ is simply a linear readout of the sensory stimuli $\vec{x}(t)$, with readout weights that are dynamically changed by the modulatory system:

$$\vec{y}(t) = (M(t) \odot W_0)\, \vec{x}(t) \tag{4}$$

where $W_0$ are the baseline weights and $M(t)$ the modulation provided by the modulatory system. $M(t)$ is of the same shape as $W_0$ and determines the element-wise multiplicative modulation of the baseline weights. Because the task requires the modulatory system to dynamically infer the context, we modelled it as a recurrent network – more specifically, a long-short-term memory network (LSTMs; *Hochreiter and Schmidhuber, 1997*) – with $N_h = 100$ hidden units. In particular, we used LSTMs with forget gates (*Gers et al., 2000*) but no peephole connections (for an overview of LSTM variants, see *Greff et al., 2017*).

In this work, we treated the LSTM as a black-box modulatory system that receives the sensory stimuli and the feedforward network's output and provides the feedback signal in return (*Figure 1a*). A linear readout of the LSTM's output determines the modulation $M(t)$ in *Equation 4*. In brief, this means that

$$M(t) = \text{LSTM}(\vec{x}(t), \vec{y}(t)),\tag{5}$$

where $\text{LSTM}(\cdot)$ is a function that returns the LSTM readout. For two-dimensional sources and sensory stimuli, for instance, $\text{LSTM}(\cdot)$ receives a concatenation of the two-dimensional vectors $\vec{x}(t)$ and $\vec{y}(t)$ as input and returns a two-by-two feedback modulation matrix – one multiplicative factor for each weight in $W_0$. The baseline weights $W_0$ were randomly drawn from the Gaussian distribution $\mathcal{N}(1, 0.001)$ and fixed throughout the task. The LSTM parameters and readout were learned during training of the model.

## Extension 1: Reducing the temporal specificity of feedback modulation

To probe our model's sensitivity to the timescale of the modulatory feedback (*Figure 2*), we added a temporal filter to *Equation 5*. In that case, the modulation $M(t)$ followed the dynamics

$$\tau \frac{dM(t)}{t} = -M(t) + \text{LSTM}(\vec{x}(t), \vec{y}(t)),\tag{6}$$

with $\tau$ being the time constant of modulation. For small $\tau$, the feedback rapidly affects the feedforward network, whereas larger $\tau$ imply a slowly changing modulatory feedback signal. The unit of this timescale is the number of samples from the source signals. Note that the timescale of the modulation should be considered relative to the timescale of the context changes $n_t$. As a default time constant, we used $\tau = 100 < n_t$ (see *Table 2*).

## Extension 2: Reducing the spatial specificity of feedback modulation

To allow for spatially diffuse feedback modulation (*Figure 3*), we added an intermediate layer between the sensory stimuli and the network output. This intermediate layer consisted of a population of $N_z = 100$ units that were modulated by the feedback, where neighbouring units were modulated similarly. More specifically, the units were arranged on a ring to allow for a spatially constrained modulation without boundary effects. The population's activity vector $\vec{z}(t)$ is described by

$$\vec{z}(t) = \vec{m}(t) \odot (W^{\text{x}} \vec{x}(t)),\tag{7}$$

with the sensory stimuli $\vec{x}(t)$, a weight matrix $W^{\text{x}}$ of size $N_z \times n_s$, and the vector of unit-specific multiplicative modulations $\vec{m}(t)$. Note that the activity of the units was not constrained to be positive here. The output of the network was then determined by a linear readout of the population activity vector according to

$$\vec{y}(t) = W^{\text{ro}} \vec{z}(t)\tag{8}$$

with a fixed readout matrix $W^{\text{ro}}$.

The modulation to a single unit   was given by

$$\tau \frac{dm_i(t)}{t} = -m_i(t) + \sum_{j=1}^{N_{\text{FB}}} K_{ij}\, l_j,\tag{9a}$$

$$\text{with} \quad l_j = \text{LSTM}(x(t), y(t))_j.\tag{9b}$$

Here, $\tau$ is the modulation time constant, $K$ is a kernel that determines the spatial specificity of modulation, $\text{LSTM}(\cdot)_j$ the $j$th feedback signal from the LSTM, and $N_{\text{FB}}$ is the total number of feedback

**Table 3.** Distributions used for randomly initialised weight parameters.

| Weights | Distribution |
| --- | --- |
| $W_0$ | $\mathcal{N}(1, 0.001)$ |
| $W^{\mathrm{x}}$ | $\mathcal{N}(0, 0.5)$ |
| $W^{\mathrm{Lx}}$ | $\mathcal{N}(0, 0.5)$ |
| $W^{\mathrm{ro}}$ | $\mathcal{N}(0, 0.5)$ |
| $W^{\mathrm{HL}}$ | $\mathcal{N}(1, 0.5) \cdot 20/N_{\mathrm{H}}$ |
| $W^{\mathrm{IL}}$ | $\mathcal{N}(1, 0.5)/N_{\mathrm{I}}$ |
| $W^{\mathrm{HI}}$ | $\mathcal{N}(1, 1) \cdot 20/N_{\mathrm{H}}$ |
| Long-short-term memory network parameters | $\mathcal{U}(-\sqrt{1/N_{\mathrm{H}}}, \sqrt{1/N_{\mathrm{H}}})$ |
| Long-short-term memory network readout | $\mathcal{U}(-\sqrt{1/N_{\mathrm{FB}}}, \sqrt{1/N_{\mathrm{FB}}})$ |

signals. As in the simple model, the $N_{\mathrm{FB}}$ feedback signals were determined by a linear readout from LSTM.

The modulation kernel $K$ was defined as a set of von Mises functions:

$$K_{ij} = \exp(\frac{1}{\sigma_m^2} \cos(z_i^{\mathrm{loc}} - l_j^{\mathrm{loc}})), \tag{10}$$

where $z_i^{\mathrm{loc}} = \frac{2\pi i}{N_z} \in [0, 2\pi]$ represents the location of the modulated unit $i$ on the ring and $l_j^{\mathrm{loc}}$ the 'preferred location' of modulatory unit $j$, that is, the location on the ring that it modulates most effectively. These 'preferred locations' $l_j^{\mathrm{loc}}$ of the feedback units were evenly distributed on the ring. The variance parameter $\sigma_m^2$ determines the spatial spread of the modulatory effect of the feedback units, that is, the spatial specificity of the modulation. Overall, the spatial distribution of the modulation was therefore determined by the number of distinct feedback signals $N_{\mathrm{FB}}$ and their spatial spread $\sigma_m^2$ (see *Table 2* for a list of network parameters).

## Extension 3: Hierarchical rate-based network

We further extended the model with spatial modulation (*Equation 7–Equation 10*) to include a two-stage hierarchy, positive rates and synaptic weights that obey Dale's law. Furthermore, we implemented the feedback modulation as a gain modulation that scales neural rates but keeps them positive. To this end, we modelled the feedforward network as a hierarchy of a lower-level and a higher-level population. Only the higher-level population received feedback modulation. Splitting the neural populations in this way allowed us to model the connections between them with weights that follow Dale's law. Furthermore, the unmodulated lower-level population could serve as a control for the emergence of context-invariant representations. The lower-level population consisted of $N_{\mathrm{L}} = 40$ rate-based neurons and the population activity vector was given by

$$\vec{z}^{\mathrm{L}}(t) = \left[ W^{\mathrm{Lx}} \vec{x}(t) \right]_+ , \tag{11}$$

where $W^{\mathrm{Lx}}$ is a fixed weight matrix, $\vec{x}(t)$ the sensory stimuli, and the rectification $[\cdot]_+ = \max(0, \cdot)$ ensures that rates are positive. The lower-level population thus provides a neural representation of the sensory stimuli. The higher-level population consisted of $N_{\mathrm{H}} = 100$ rate-based neurons that received feedforward input from the lower-level population. The feedforward input consisted of direct excitatory projections as well as feedforward inhibition through a population of $N_{\mathrm{I}} = 20$ local inhibitory neurons. The activity vector of the higher-level population $\vec{z}^{\mathrm{H}}(t)$ was thus given by

$$\vec{z}^{\mathrm{H}}(t) = \left[ \vec{p}(t) \odot (W^{\mathrm{HL}} \vec{z}^{\mathrm{L}}(t) - W^{\mathrm{HI}} \vec{z}^{\mathrm{I}}(t)) \right]_+ \tag{12}$$

$$\vec{z}^{\mathrm{I}}(t) = \left[ W^{\mathrm{IL}} \vec{z}^{\mathrm{L}}(t) \right]_{+} \quad . \tag{13}$$

Here, $W^{\mathrm{HL}}$, $W^{\mathrm{HI}}$, and $W^{\mathrm{IL}}$ are positive weight matrices, $\vec{z}^{\mathrm{I}}(t)$ is the inhibitory neuron activities, and $\vec{p}(t)$ is the neuron-specific gain modulation factors. As for the spatially modulated network of Extension 2, the network output $\vec{y}(t)$ was determined by a fixed linear readout $W^{\mathrm{ro}}$ (see **Equation 8**). The distributions used to randomly initialise the weight matrices are provided in **Table 3**.

Again, the modulation was driven by feedback from the LSTM, but in this model variant we assumed inhibitory feedback, that is, stronger feedback signals monotonically decreased the gain. More specifically, we assumed that the feedback signal targets a population of modulation units $\vec{m}$, which in turn modulate the gain in the higher-level population. The gain modulation of neuron was constrained between 0 and 1 and determined by

$$p_i(t) = \frac{1}{1 + \exp(m_i(t))} \tag{14}$$

with $m_i(t)$ being the activity of a modulation unit , which follows the same dynamics as in **Equation 9a** (see **Figure 6a**).

## Training the model

We used gradient descent to find the model parameters that minimise the difference between the source signal $\vec{s}(t)$ and the feedforward network's output $\vec{y}(t)$:

$$\mathcal{L} = \sum_{t=1}^{n_t} \mathrm{dist}(\vec{s}(t), \vec{y}(t)) \tag{15}$$

with a distance measure $\mathrm{dist}(\cdot)$. We used the machine learning framework PyTorch (**Paszke et al., 2019**) to simulate the network model, obtain the gradients of the objective $\mathcal{L}$ by automatic differentiation, and update the parameters of the LSTM using the Adam optimiser (**Kingma and Ba, 2014**) with a learning rate of $\eta = 10^{-3}$. As distance measure in the objective, we used a smooth variant of the L1 norm (PyTorch's smooth L1 loss variant) because it is less sensitive to outliers than the mean squared error (**Huber, 1964**).

During training, we simulated the network dynamics over batches of 32 trials using forward Euler with a time step of $\Delta t = 1$. Each trial consisted of $n_t$ time steps (i.e. samples) and the context (i.e. mixing matrix) differed between trials. Since the model contains feedback and recurrent connections, we trained it using backpropagation through time (**Werbos, 1990**). This means that for each trial we simulated the model and computed the loss for every time step. At the end of the trial, we propagated the error through the $n_t$ steps of the model to obtain the gradients and updated the parameters accordingly (**Figure 8**). Although the source signals were the same in every trial, we varied their phase independently across trials to prevent the LSTM from learning the exact signal sequence. To this end, we generated 16,000 samples of the source signals and in every batch randomly selected chunks of $n_t$ samples independently from each source. Model parameters were initialised according to the distributions listed in **Table 3**.

In all model variants, we optimised the parameters of the modulator (input, recurrent, and readout weights as well as the biases of the LSTM; see **Equation 5** and **Equation 9b**). The parameters were initialised with the defaults from the corresponding PyTorch modules, as listed in **Table 3**. To facilitate the training in the hierarchical rate-based network despite additional constraints, we also optimised the feedforward weights $W^{\mathrm{HL}}$, $W^{\mathrm{HI}}$, $W^{\mathrm{IL}}$, $W^{\mathrm{Lx}}$, and $W^{\mathrm{ro}}$. In principle, this allows to adapt the representation in the two intermediate layers such that the modulation is most effective. However, although we did not quantify it, we observed that optimising the network readout $W^{\mathrm{ro}}$ facilitated the training the most, suggesting that a specific format of the sensory representations was not required for an effective modulation.

To prevent the gain modulation factor from saturating at 0 or 1, we added a regularisation term $\mathcal{R}$ to the loss function **Equation 15** that keeps the LSTM's output small:

$$\mathcal{R} = \lambda_{\text{out}} \sum_{t=1}^{n_t} \sum_{j=1}^{N_{\text{FB}}} \left| \text{LSTM}(x(t), y(t))_j \right| \tag{16}$$

with $\lambda_{\text{out}} = 10^{-5}$.

Gradient values were clipped between –1 and 1 before each update to avoid large updates. For weights that were constrained to be positive, we used their absolute value in the model. Each network was trained for 10,000–12,000 batches and for 5 random initialisations (***Figure 1—figure supplement 2***).

## Testing and manipulating the model

We tested the network model performance on an independent random set of contexts (i.e. mixing matrices), but with the same source signals as during training. During testing, we also changed the context every $n_t$ steps, but the length of this interval was not crucial for performance (***Figure 1—figure supplement 1d***).

To manipulate the feedback modulation in the hierarchical rate-based network (***Figure 4***), we provided an additional input to the modulation units $m$ in ***Equation 9a***. We used an input of 3 or –3 depending on whether the modulation units were activated or inactivated, respectively. To freeze the feedback modulation (***Figure 6***), we discarded the feedback signal and held the local modulation $p$ in ***Equation 14*** at a constant value determined by the feedback before the manipulation. The dynamics of the LSTM were continued, but remained hidden to the feedforward network until the freezing was stopped.

## Unmodulated feedforward network models

### Linear regression

As a control, we trained feedforward networks with weights that were not changed by a modulatory system. First, we used the simplest possible network architecture, in which the sensory stimuli are linearly mapped to the outputs (***Figure 1—figure supplement 1a***):

$$y(t) = Wx(t). \tag{17}$$

It is intuitive that a fixed set of weights $W$ cannot invert two different contexts (i.e. different mixing matrices $A_1$ and $A_2$). As an illustration, we trained this simple feedforward network on one context and tested it on different contexts. To find the weights $W$, we used linear regression to minimise the mean squared error between the source signal $s(t)$ and the network's output $y(t)$. The training data consisted of 1024 consecutive time steps of the sensory stimuli for a fixed context, and the test data consisted of different 1024 time steps generated under a potentially different mixing. We repeated this procedure by training and testing a network for all combinations of 20 random contexts.

### Multilayer nonlinear network

Since solving the task was not possible with a single set of readout weights, we extended the feedforward model to include three hidden layers consisting of 32, 16, and 8 rectified linear units (***Figure 1—figure supplement 1d***). The input to this network was one time point from the sensory stimuli and the target output the corresponding time point of the sources. We trained the multilayer network on 5000 batches of 32 contexts using Adam (learning rate 0.001) to minimise the mean squared error between the network output and the sources.

### Multilayer network with sequences as input

Solving the task requires the network to map the same sensory stimulus to different outputs depending on the context. However, inferring the context takes more than one time point. To test if a feedforward network with access to multiple time points at once could in principle solve the task, we changed the architecture of the multilayer network, such that it receives a sequence of the sensory stimuli (***Figure 1—figure supplement 1g***). The output of the network was a sequence of equal length. We again trained this network on 5000 batches of 32 contexts to minimise the error between its output and the target sources, where both the network input and output were sequences. The length of these sequences was varied between 1 and 150.

## Data analysis
### Signal clarity
To determine task performance, we measured how clear the representation of the source signals is in the network output. We first computed the correlation coefficient of each signal $s_i$ with each output $y_j$

$$r_{ij} = \frac{\sum_t (s_i(t) - \bar{s}_i)(y_j(t) - \bar{y}_j)}{\sigma_{s,i}\sigma_{y,j}} \quad , \tag{18}$$

where $\bar{s}_i$ and $\bar{y}_j$ are the respective temporal mean and $\sigma_{s,i}$ and $\sigma_{y,j}$ the respective temporal standard deviations. The signal clarity in output $y_j$ is then given by the absolute difference between the absolute correlation with one compared to the other signal:

$$c_j = ||r_{1j}| - |r_{2j}|| \quad . \tag{19}$$

By averaging over outputs, we determined the overall signal clarity within the output. Note that the same measure can be computed on other processing stages of the feedforward network. For instance, we used the signal clarity of sources in the sensory stimuli as a baseline control.

### Signal-to-noise ratio
The signal-to-noise ratio in the sensory stimuli was determined as the variability in the signal compared to the noise. Since the mean of both the stimuli and the noise was zero, the signal-to-noise ratio could be computed by

$$\text{SNR} = \frac{\sigma_s^2}{\sigma_n^2} \quad ,$$

where $\sigma_n$ is the standard deviation of the additive white noise and $\sigma_s$ is the measured standard deviation in the noise-free sensory stimuli, which was around 0.32. As a scale of the signal-to-noise ratio, we used decibels (dB), that is, we used $\text{dB} = 10\log_{10}(\text{SNR})$.

## Linear decoding analysis
### Signal decoding
We investigated the population-level invariance by using a linear decoding approach. If there was an invariant population subspace, the source signals could be decoded by the same decoder across different contexts. We therefore performed linear regression between the activity in a particular population and the source signals. This linear decoder was trained on $n_c = 10$ different contexts with $n_t = 1,000$ time points each, such that the total number of samples was 10,000. The linear decoding was then tested on 10 new contexts and the performance determined using the $R^2$ measure.

### Context decoding
We took a similar approach to determine from which populations the context could be decoded. For the dynamic blind source separation task, the context is given by the source mixture, as determined by the mixing matrix. Since we normalised the rows of each mixing matrix, the context was determined by two context variables. We calculated the temporal average of the neuronal activities within each context and performed a linear regression of the context variables onto these averages. To exclude onset transients, we only considered the second half (500 samples) of every context. Contexts were sampled from the two-dimensional grid of potential contexts. More specifically, we sampled 20 points along each dimension and excluded contexts, in which the sensory stimuli were too similar (analogously to the generation of mixing matrices), leaving 272 different contexts (see *Figure 7c*, right). The linear decoding performance was determined with a fivefold cross-validation and measured using $R^2$. Since the modulatory feedback signals depend nonlinearly on the context (*Figure 7c*), we tested two nonlinear versions of the decoding approach. First, we performed a quadratic expansion of the averaged population activity before a linear decoding. Second, we tested a linear decoding of the inverse mixing matrix (four weights) instead of the two variables determining the context.

## Population subspace analysis

We visualised the invariant population subspaces by projecting the activity vector onto the two readout dimensions and the first principal component. To measure how the orientation of the subspaces changes when the context or feedback changes, we computed the angle between the planes spanned by the respective subspaces. These planes were fitted on the three-dimensional data described above using the least-squares method. Since we were only interested in the relative orientation of the subspaces, we used a circular measure of the angles, such that a rotation of 180° corresponded to 0°. This means that angles could range between 0 and 90°.

## Code availability

The code for models and data analysis is publicly available under https://github.com/sprekelerlab/feedback_modulation_Naumann22, (copy archived at swh:1:rev:05373b093803e464082ad5b9e-8ab2dbbf43bb23e; *Naumann, 2022*).

## Acknowledgements

We thank Owen Mackwood for providing a code framework that manages simulations on a compute cluster, Loreen Hertäg and Johannes Letzkus for feedback on the manuscript, and the members of the Sprekeler lab for valuable discussions. No external funding was received for this work.

# Additional information

### Funding

No external funding was received for this work.

### Author contributions

Laura B Naumann, Conceptualization, Formal analysis, Investigation, Methodology, Project administration, Software, Visualization, Writing – original draft, Writing – review and editing; Joram Keijser, Conceptualization, Methodology, Project administration, Supervision, Writing – original draft, Writing – review and editing; Henning Sprekeler, Conceptualization, Funding acquisition, Methodology, Project administration, Resources, Supervision, Writing – original draft, Writing – review and editing

### Author ORCIDs

Laura B Naumann ⓘ http://orcid.org/0000-0002-7919-7349
Henning Sprekeler ⓘ http://orcid.org/0000-0003-0690-3553

### Decision letter and Author response

Decision letter https://doi.org/10.7554/eLife.76096.sa1
Author response https://doi.org/10.7554/eLife.76096.sa2

# Additional files

### Supplementary files

• MDAR checklist

### Data availability

The current manuscript is a computational study, so no data have been generated for this manuscript. Modelling code is available under https://github.com/sprekelerlab/feedback_modulation_Naumann21, (copy archived at swh:1:rev:05373b093803e464082ad5b9e8ab2dbbf43bb23e) upon publication.

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
