## [Editor Report]

One of the key questions in sensory neuroscience is how cortical networks extract invariant percepts from variable sensory inputs. While much of the literature focuses on the role of feedforward hierarchical processing for extracting invariant percepts, this study proposes a novel implementation based on top-down feedback. The article analyses the underlying mechanism based on an invariant subspace and presents instantiations of this mechanism at different levels of biophysical realism.

---

## [Decision Letter]

**Decision letter after peer review:**

[Editors’ note: the authors submitted for reconsideration following the decision after peer review. What follows is the decision letter after the first round of review.]

Thank you for submitting the paper "Invariant neural subspaces maintained by feedback modulation" for consideration by *eLife*. Your article has been reviewed by 3 peer reviewers, one of whom is a member of our Board of Reviewing Editors, and the evaluation has been overseen by a Senior Editor. The reviewers have opted to remain anonymous.

We are sorry to say that, after consultation with the reviewers, we have decided that this work will not be considered further for publication by *eLife* at this time.

While all three reviewers appreciated the novelty of the proposed computational role for feedback connections, they estimated that substantial additional work would be needed to establish more firmly the mechanisms underlying context–invariant processing and its biological relevance. Given the extent of the criticisms, we have decided to reject the paper. Should further analyses allow you to fully address these criticisms we would be open to a resubmission.

*Reviewer #1:*

One of the key questions in sensory neuroscience is how cortical networks extract invariant percepts from variable sensory inputs. Much of the existing literature focuses on the role of feed–forward hierarchical processing for extracting such invariances. The present study proposes an alternative mechanism based on top–down feedback. Focusing on the so–called source–separation, or cocktail–party problem, the manuscript shows how sources mixed in a context–dependent manner can be separated independently of context, using feed–forward networks modulated by top–down context–dependent inputs.

The manuscript starts with a simplified, abstract network, and then progressively moves to more biologically plausible ones. By performing population analyses of network activity, the authors then argue for a mechanism based on context–invariant subspaces.

Strengths of the paper:

– novel proposal for an important class of cortical computations

– very elegant formulation of the problem

– the writing style is very clear and appealing

– network implementations at different levels of biophysical realism.

Weaknesses of the paper:

– the announced mechanism, based on invariant subspaces, is not clearly explained and needs to be supported by additional evidence.

– how the network detects contextual changes does not seem to be explained

– the analyses of network activity, their rationale and the resulting conclusions are difficult to follow.

While I very much appreciated the novelty and the elegance of the approach developed in this paper, ultimately, I was left wondering how the networks perform their task.

– The title and abstract announce a mechanism based on invariant neural subspaces. Clearly, since the readout is fixed, there must be an invariant subspace, but the key question is how it is generated and maintained across contexts. In the Results, this mechanism is explained only briefly at the very end of the results, in connection to Figure 6, which seems to be just an illustration. The authors would need to unpack what precisely the mechanism is (not clear right now) and give more evidence for it.

– An important complementary issue is how the network detects context changes. The manuscript states that "feedback–mediated invariance requires time to establish after contextual changes" (lines 245–246), but how this works does not seem to be explained. What type of error signal does the network use to change the gains?

On a related note, is the network trained on all the contexts it sees during testing, or is it able to deal with totally novel contexts?

– The logic of the sequence of analysis (optogenetic manipulations; correlation; changes in gain…) is a bit difficult to follow and needs more motivation. In particular, why is the non–linear encoding of context important?

– It is a bit surprising that the analyses focus on the most complex version of the network to examine mechanisms. Presumably the simplified networks could be leveraged to identify and explain the mechanisms in a more transparent manner.

*Reviewer #2:*

The authors aim to explore an understudied potential function of feedback connections: providing context–independent sensory processing. Invariant sensory processing is frequently assumed to be carried out by feedforward processing and much of the study of feedback focuses on how feedback could implement context–dependent processing. This makes this study promising and relatively novel.

The strengths of this paper are that it demonstrates convincingly and using a variety of network architectures and feedback mechanisms that feedback modulations can indeed help a network read out sensory input in a context–independent way.

The weaknesses are in the analysis and comparisons of the various networks. While the basic finding that this invariance does not result from invariant activity on the individual neuron level is interesting and of value, the explanation that it instead leads to invariant population activity is almost tautological given the network architecture. It is also unclear how the simpler models the authors present are meant to provide insight on either the more biologically detailed hierarchical model or on real neural processing, especially given that the mode of modulation in the simplest model (re–weighting of feedforward weights) differs from that of the later models (re–weighting of neural activation). In this way I don't feel that the authors fully achieved their goal of describing the mechanism of feedback modulation.

The methods appear technically sound, but I am confused by some of the choices. For example, the authors start with a single layer network where feedback modulates the weights between the input and output. This is a different mechanism than the normal neuronal gain usually attributed to feedback. The authors then add more details to push the model more in the biological direction, but multiple details are sometimes added at once and the logic behind these choices isn't always clear. I believe the authors switch to using neuronal gain when they want to explore spatially correlated modulation, but they don't talk about neuronal modulation until they introduce their full hierarchical model. The hierarchical model also adds Dale's law and a separate inhibitory population but it is not clear why these details were added or if/how they change the function of the model in a way relevant to understanding feedback modulation. Even the use of a multi–layer model is not very well motivated given that they show that this task can be completed with a very small one layer model. The simplicity of the task has implications for understanding some of these findings as well. For example, to show that modulatory signals can be spatially correlated, the authors create a model with many more neurons than is needed to solve the task and show that the modulatory signal can target nearby cells in this population similarly without sacrificing performance. But the low dimensional nature of the modulatory signal is only really an issue of interest in the context of a higher dimensional task. As a thought experiment: if the 2 neurons in the original model were simply replicated to 50 each and each population of 50 neurons was given the same modulation, this would be essentially equivalent to the original 2 cell model, but under the logic of what the authors have shown here, would supposedly demonstrate that modulatory signals still work if low dimensional. In this way, that analysis fell short.

I think that this work may spur more interest in studying the role of feedback for invariant sensory processing, which would be a very productive outcome. Furthermore, the demonstration that the context signals cannot be linearly readout from the cells performing the modulation is an important lesson for the analysis of neural data. I also think further reflection on the finding that the modulatory network needs direct sensory input (more so even than the input from later processing stages) will be very important for understanding how this modulation works and how it relates to biological structures. As the authors note, this may mean that their model is more akin to inputs from higher order thalamic areas, though even that mapping is imperfect due to the lack of recurrence.

I think it would help the readability of the paper if the authors included a few more brief descriptions of the methods in the Results. For example, a better description of how the signals are generated, the fact that the networks are trained with a single set of signals only, etc. Also, there were points where it wasn't clear if a network was tested under different conditions or actually retrained for them (for example, in figure 2d/e). Also, the fact that the modulation went from being on the weights to on the neurons themselves was not made clear in section "Invariance can be established by spatially diffuse feedback modulation". I also found the schematic in Figure 1a a bit confusing. I don't know why x is represented as a question mark when it is a sum of the two signals. I'd prefer a diagram that makes the dimensionality of x clearer (relatedly, why are there only 3 weights from x to y when I believe it is a 2x2 matrix).

"While we trained the modulatory system using supervised learning, the contextual inference is performed by its dynamics without access to the target sources and thus unsupervised" I feel this could be read as saying that an actual unsupervised objective was used, when in fact only supervised learning took place, so I would suggest re–wording.

I didn't understand the claim about matched EI inputs and how it depends on using gain modulation. This should probably be expanded and related to the main questions of the paper or possibly removed.

Figure 4i seems to be the main demonstration that individual neural activity itself is not invariant to context. I'd like to see a more in–depth exploration of this. Particularly, if the readout only relied on a small handful of neurons then finding that the rest of the neurons are not context–invariant wouldn't prove that individual neural invariance is not a relevant mechanism. Given that the readout from this network is known, it would be particularly easy to determine if the heavily weighted neurons in particular are or are not context invariant.

In general, I don't understand why the authors use a separately trained linear readout when trying to show that the population activity at the final layer is invariant. They eventually acknowledge that "Since this readout is obtained from the data, this procedure does not require knowledge of the readout in the network model. Note that the trained decoder and the network readout are not necessarily identical" but they don't explain why they are using this alternative readout or what new insights its use adds. Particularly, the performance of the network indicates the there is some sort of context invariant read out possible from this population, yet the authors use this other readout in a way that is seemingly supposed to add something to the explanation.

Be sure to say what errorbars are based on in all figures.

"In our model, the mechanism needs to satisfy a few key

requirements: i) the modulation is not uniform across the population, ii) it operates on a timescale similar to that of changes in context, and iii) it is driven by feedback projections." I don't understand claim (iii). If anything, the results show the importance of the modulation being driven by feedforward sensory signals (figure 2d/e).

"In addition, feedback inputs from the sensory to the modulatory system allow a better control of the modulated network state." I don't see how the connections from a sensory system to a modulatory system are "feedback".

*Reviewer #3:*

I appreciate the didactic way in which the manuscript was written (and beautiful figures!), in particular the progression from a vanilla architecture towards the full fledged model with EI rectified neurons with spatially specific modulation. My main concerns (detailed below) are two–fold:

1. I felt that some extensions were not explicitly justified (e.g. why 2 layers instead of 2, etc)

2. I was expecting more 'reverse–engineering' of the mechanism through which the network accomplishes a context invariant projection. This is the main result of the paper, as reflected in the title, so I think it deserves more unpacking. Below I unpack these concerns, sometimes providing some suggestions to improve the motivation and clarity of the paper (without any particular order)

1. Overall, the architecture choices are a bit unjustified. In the extreme, wouldn't the LSTM alone solve the task? The addition of each feedforward layer should be better motivated (e.g. more biologically realistic? In what sense?). For example, why add an extra layer from extensions 2 and 3? If those are necessary, this should be explained. If they are not necessary, they should be removed.

2. 'Because the task requires a dynamic inference of the context, it cannot be solved by feedforward networks or standard blind source separation algorithms' I think the paper could be better motivated if this was shown explicitly with some examples.

3. A figure explicitly illustrating the training setup would help motivate what is trivially solved and what is actually challenging. For instance, in the main manuscript, it is not clear in which cases the network is trained and tested on the same contexts (ie A(t)) and which cases it is not. In the first case, the context can be easily inferred from x(t) but the latter is more challenging?

4. however I understand that the paper is already too long, Intra / extrapolation results deserve more spotlight and unpacking in my opinion. In general, if there is a lack of space, I would merge Figure 1 and figure 2 – and jump directly to extension 1 – and move most of figure 2 to sup.

5. Most important concern to me: Figure 6, in which the mechanism is revealed, deserves more quantifications to explicitly pinpoint the mechanism. Three suggestions come to mind:

a. Plot the 3 PCs components (instead of just 1) and show the readout in this space. The key result is that the readout is invariant to context and this is not clearly illustrated at the moment. Instead, what is shown is that the representation changes, but that it changes in a way that preserves invariance on the readout is not clearly highlighted.

b. The authors highlight that the network is not just reversing the new mixing coefficients and projecting the activity back into the 2d low manifold. Instead, it is rotating everything out of this manifold. My suggestion would be to show this alternatively explicitly. Is it actually possible? Relatedly, what happens if the context is changed back to context 1?

c. Finally, all the statements made about this figure should be quantified and not just illustrated for 1 trial.

---

## [Author Response]

[Editors’ note: The authors appealed the original decision. What follows is the authors’ response to the first round of review.]

Reviewer #1:One of the key questions in sensory neuroscience is how cortical networks extract invariant percepts from variable sensory inputs. Much of the existing literature focuses on the role of feed–forward hierarchical processing for extracting such invariances. The present study proposes an alternative mechanism based on top–down feedback. Focusing on the so–called source–separation, or cocktail–party problem, the manuscript shows how sources mixed in a context–dependent manner can be separated independently of context, using feed–forward networks modulated by top–down context–dependent inputs.The manuscript starts with a simplified, abstract network, and then progressively moves to more biologically plausible ones. By performing population analyses of network activity, the authors then argue for a mechanism based on context–invariant subspaces.Strengths of the paper:– novel proposal for an important class of cortical computations– very elegant formulation of the problem– the writing style is very clear and appealing– network implementations at different levels of biophysical realism.Weaknesses of the paper:– the announced mechanism, based on invariant subspaces, is not clearly explained and needs to be supported by additional evidence.– how the network detects contextual changes does not seem to be explained– the analyses of network activity, their rationale and the resulting conclusions are difficult to follow.

We thank the reviewer for their interest in our work. We hope to have addressed all of the weaknesses listed above with the revision of the manuscript. In brief:

– We provide a more in-depth analysis of the mechanism based on invariant subspaces (see Change 2 above).

– We show and discuss what the network needs to detect context changes (see Change 4 above).

– We rearranged the results in the manuscript and added additional explanations to make the sequence of analyses and main findings clearer (see Change 1 and 3 above).

While I very much appreciated the novelty and the elegance of the approach developed in this paper, ultimately, I was left wondering how the networks perform their task.– The title and abstract announce a mechanism based on invariant neural subspaces. Clearly, since the readout is fixed, there must be an invariant subspace, but the key question is how it is generated and maintained across contexts. In the Results, this mechanism is explained only briefly at the very end of the results, in connection to Figure 6, which seems to be just an illustration. The authors would need to unpack what precisely the mechanism is (not clear right now) and give more evidence for it.

We agree with the reviewer that the mechanism based on invariant subspaces needed more unpacking to underscore the main claims of the paper. As outlined in Change 2 above, we have extended our analyses of the mechanism and provided a quantification of the findings illustrated in the previous Figure 6.

– An important complementary issue is how the network detects context changes. The manuscript states that "feedback–mediated invariance requires time to establish after contextual changes" (lines 245–246), but how this works does not seem to be explained. What type of error signal does the network use to change the gains?On a related note, is the network trained on all the contexts it sees during testing, or is it able to deal with totally novel contexts?

We thank the reviewer for pointing out that this was not sufficiently explained. As described in Change 4, we have added several sentences and a supplementary figure to clarify how the modulator maps its input to the correct feedback modulation:

“[…] The modulator therefore had to use its recurrent dynamics to determine the appropriate modulatory feedback for the time-varying context, based on the sensory stimuli and the network output. Put differently, the modulator had to learn an internal model of the sensory data and the contexts, and use it to establish the desired context invariance in the output.“ (ll. 57-61)

“Because the network had to learn an internal model of the task, we expected a limited degree of generalisation to new situations. Indeed, the network was able to interpolate between source frequencies seen during training (Supp. Figure S5), but failed on sources and contexts that were qualitatively different (Supp. Figure S6 b-d). The specific computations performed by the modulator are therefore idiosyncratic to the problem at hand. Hence, we did not investigate the internal dynamics of the modulator in detail, but concentrated on its effect on the feedforward network.” (ll. 85-91)

These excerpts should also answer the question regarding new contexts. Note that contexts were randomly sampled from a continuous context space for training and testing. The network is therefore not tested on the exact same contexts, but on contexts from the same distribution (unless specified otherwise). We have also clarified this in the methods section:

“Unless specified otherwise, we sampled new contexts for each training batch and for the test data, such that the training and test data followed the same distribution without necessarily being the same.” (ll. 494-497)

Regarding the question of the error signal: The modulator does not use an error signal, but computes a new mapping from its input to the correct modulation in response to context changes. Since the inputs are time-dependent the modulator needs to see a sufficient number of time-points before it can provide the appropriate feedback signal.

In additional analyses we found that a feedforward network can also solve the task, if it is given a time window of the sensory signals rather than a single moment in time. Such a network requires about the same number of timesteps from the stimuli (compare with Supp. Figure S1i and Supp. Figure S6a) as the modular-based architecture. This shows that the time it takes the network to infer the context from its input is not particular to our model but to the task, i.e. the statistics of the sources and contexts.

– The logic of the sequence of analysis (optogenetic manipulations; correlation; changes in gain…) is a bit difficult to follow and needs more motivation. In particular, why is the non–linear encoding of context important?

As described in the general answer, we agree that the sequence of analyses was not intuitive to the reader and thus rearranged the results in the new version of the manuscript. In particular, we have removed the optogenetic manipulations and considerations of E/I balance, since they are not central to the message of the paper (see Change 1 and 3 above).

Regarding the non-linear encoding of the context: We now include a deeper discussion of the results (see Change 3) that aim to clarify the relevance of these findings:

“In summary, the higher-population provides a linear representation not only of the stimuli, but also of the context. In contrast, the modulatory units contained a nonlinear representation of the context, which could not be extracted by linear decoding approaches. We speculate that if contextual feedback modulation is mediated by interneurons in layer 1, they should represent the context in a nonlinear way.” (ll. 288-292).

– It is a bit surprising that the analyses focus on the most complex version of the network to examine mechanisms. Presumably the simplified networks could be leveraged to identify and explain the mechanisms in a more transparent manner.

We want to thank the reviewer for this suggestion as we think it has significantly helped us to improve the structure of the manuscript. We now perform the single cell and population analyses on the network with spatially diffuse modulation (from Figure 3), as this is the simplest model comprising a neural population. We then verify that the findings hold for the Dalean network (new Figure 6). Details on the new order of the results can be found in the list of major changes above (specifically point 1).

Reviewer #2:The authors aim to explore an understudied potential function of feedback connections: providing context–independent sensory processing. Invariant sensory processing is frequently assumed to be carried out by feedforward processing and much of the study of feedback focuses on how feedback could implement context–dependent processing. This makes this study promising and relatively novel.The strengths of this paper are that it demonstrates convincingly and using a variety of network architectures and feedback mechanisms that feedback modulations can indeed help a network read out sensory input in a context–independent way.The weaknesses are in the analysis and comparisons of the various networks. While the basic finding that this invariance does not result from invariant activity on the individual neuron level is interesting and of value, the explanation that it instead leads to invariant population activity is almost tautological given the network architecture. It is also unclear how the simpler models the authors present are meant to provide insight on either the more biologically detailed hierarchical model or on real neural processing, especially given that the mode of modulation in the simplest model (re–weighting of feedforward weights) differs from that of the later models (re–weighting of neural activation). In this way I don't feel that the authors fully achieved their goal of describing the mechanism of feedback modulation.The methods appear technically sound, but I am confused by some of the choices. For example, the authors start with a single layer network where feedback modulates the weights between the input and output. This is a different mechanism than the normal neuronal gain usually attributed to feedback. The authors then add more details to push the model more in the biological direction, but multiple details are sometimes added at once and the logic behind these choices isn't always clear. I believe the authors switch to using neuronal gain when they want to explore spatially correlated modulation, but they don't talk about neuronal modulation until they introduce their full hierarchical model. The hierarchical model also adds Dale's law and a separate inhibitory population but it is not clear why these details were added or if/how they change the function of the model in a way relevant to understanding feedback modulation. Even the use of a multi–layer model is not very well motivated given that they show that this task can be completed with a very small one layer model. The simplicity of the task has implications for understanding some of these findings as well. For example, to show that modulatory signals can be spatially correlated, the authors create a model with many more neurons than is needed to solve the task and show that the modulatory signal can target nearby cells in this population similarly without sacrificing performance. But the low dimensional nature of the modulatory signal is only really an issue of interest in the context of a higher dimensional task. As a thought experiment: if the 2 neurons in the original model were simply replicated to 50 each and each population of 50 neurons was given the same modulation, this would be essentially equivalent to the original 2 cell model, but under the logic of what the authors have shown here, would supposedly demonstrate that modulatory signals still work if low dimensional. In this way, that analysis fell short.I think that this work may spur more interest in studying the role of feedback for invariant sensory processing, which would be a very productive outcome. Furthermore, the demonstration that the context signals cannot be linearly readout from the cells performing the modulation is an important lesson for the analysis of neural data. I also think further reflection on the finding that the modulatory network needs direct sensory input (more so even than the input from later processing stages) will be very important for understanding how this modulation works and how it relates to biological structures. As the authors note, this may mean that their model is more akin to inputs from higher order thalamic areas, though even that mapping is imperfect due to the lack of recurrence.

We thank the reviewer for their thorough assessment of our work and for raising important concerns about the sequence of analyses and line of argumentation. In the following we first address what we think are the main concerns raised. After that, we respond to the specific recommendations in more detail.

Main concerns and answers:

1. Concern: The emergence of subspaces at the population level is not a surprising finding.

It is indeed not surprising that the neural population contains a context-invariant subspace. We do not think the existence of an invariant subspace is the main finding of our work, but rather how this subspace can be dynamically maintained by feedback modulation. Nevertheless, we think it is worth noting that the invariant subspace can not only be extracted with the readout learned during training, but also with a separate decoder that was trained on only a few contexts. To make clearer why this could be of interest, we have edited the respective text:

“In fact, the population had to contain an invariant subspace, because the fixed linear readout of the population was able to extract the sources across contexts. However, the linear decoding approach shows that this subspace can be revealed from the population activity itself with only a few contexts and no knowledge of how the neural representation is used downstream. The same approach could therefore be used to reveal context-invariant subspaces in neural data from population recordings.” (ll. 179-185)

2. Concern: It is unclear how simpler models give insight into the more complex models. Relatedly, some model choices are not well motivated.

We acknowledge that the connection between the models was not made sufficiently clear. To address this, we made several changes to the manuscript. First, we demonstrate explicitly that the network with spatial modulation finds an equivalent solution to the initial linear readout network. In particular, we show that the effective weights from the stimuli to the network output in the network with spatial modulation also follow the inverse of the mixing (Supp. Figure S8d) and describe this in the text:

“The diffuse feedback modulation switched when the context changed, but was roughly constant within contexts (Figure 3c), as in the simple model. The effective weight from the stimuli to the network output also inverted the linear mixture of the sources (Supp. Figure S8d, compare with Figure 1c).” (ll. 143-146)

Second, we perform the population level analyses on the simpler spatially modulated model and then verify that the same results hold for the Dalean network (see Major Change 1 and 3 above). This demonstrates that our findings are not a result of the specific architecture of the feedforward model. Third, we have added a few more sentences to motivate the distinction between the two neural populations in the Dalean model:

“We extended the feedforward model as follows (Figure 6a): First, all neurons had positive firing rates. Second, we split the neural population (z in the previous model) into a "lower-level" (z^L^) and "higher-level" population (z^H^). The lower-level population served as a neural representation of the sensory stimuli, whereas the higher-level population was modulated by feedback. This allowed a direct comparison between a modulated and an unmodulated neural population. It also allowed us to include Dalean weights between the *two populations.” (ll. 236-242)*

3. Concern: The feedback can only be low-dimensional, because the task is low-dimensional.

The reviewer is right that the low dimensionality of the task is why the feedback modulation can be low-dimensional. More precisely, it is the low dimensionality of the context that allows a low-dimensional feedback modulation. We acknowledge this in lines 148-150 of the manuscript:

“Moreover, the feedback could have a spatially broad effect on the modulated population without degrading the signal clarity (Figure 3e, Supp. Figure S6), consistent with the low dimensionality of the context.” (ll. 148-150)

The size of the neural populations that are necessary to solve the task are more related to the dimensionality of the stimuli or the degree of nonlinearity in the input-output mapping. Therefore, a low-dimensional and diffuse modulation may still be functional for more high-dimensional or nonlinear tasks, as long as the context remains low-dimensional. We also think that for higher dimensional inputs feedforward mechanisms could play a role in preprocessing them either directly towards invariance or into a form where the modulation can achieve it most effectively.

I think it would help the readability of the paper if the authors included a few more brief descriptions of the methods in the Results. For example, a better description of how the signals are generated, the fact that the networks are trained with a single set of signals only, etc. Also, there were points where it wasn't clear if a network was tested under different conditions or actually retrained for them (for example, in figure 2d/e). Also, the fact that the modulation went from being on the weights to on the neurons themselves was not made clear in section "Invariance can be established by spatially diffuse feedback modulation". I also found the schematic in Figure 1a a bit confusing. I don't know why x is represented as a question mark when it is a sum of the two signals. I'd prefer a diagram that makes the dimensionality of x clearer (relatedly, why are there only 3 weights from x to y when I believe it is a 2x2 matrix).

We thank the reviewer for providing specific recommendations on how to improve the readability of the manuscript. We have implemented this advice by adding the key equations of the setup to the results. We also added a methods figure that explicitly illustrates the task, the model and the training setup on a more technical level (see Figure 8). Furthermore, we adapted Figure 1a according to the reviewer’s recommendation.

Regarding the lack of clarity if the network was retrained or not, we hope that the additional analyses on the generalisation ability of the network help make this point clearer. To make explicit that networks were retrained for Figure 2 we modified the text to say:

“To investigate how the timescale of modulation affects the performance in the dynamic blind source separation task, we trained network models, in which the modulatory feedback had an intrinsic timescale that forced it to be slow.” (ll. 103-105)

Finally, it is correct that in Figure 3 the modulation went from modulating weights to modulating neurons. For the spatially diffuse modulation, we assumed that all weights to a neuron receive the same modulation. Since synaptic inputs are integrated linearly, this is equivalent to a neuronal gain modulation. We have added an explicit explanation to the results:

“We here assume that all synaptic weights to a neuron receive the same modulation, such that the feedback performs a gain modulation of neural activity (Ferguson and Cardin, 2020).” (ll. 137-139)

"While we trained the modulatory system using supervised learning, the contextual inference is performed by its dynamics without access to the target sources and thus unsupervised" I feel this could be read as saying that an actual unsupervised objective was used, when in fact only supervised learning took place, so I would suggest re–wording.

Good point. We have changed this sentence to make clear that the training of the network itself is unsupervised. It now reads:

“The modulator was trained using supervised learning. Afterwards, its weights were fixed and it no longer had access to the target sources (see Materials and methods, Figure 8). The modulator therefore had to use its recurrent dynamics to determine the appropriate modulatory feedback for the time-varying context, based on the sensory stimuli and the network output. Put differently, the modulator had to learn an internal model of the sensory data and the contexts, and use it to establish the desired context invariance in the output.” (ll. 55-61)

I didn't understand the claim about matched EI inputs and how it depends on using gain modulation. This should probably be expanded and related to the main questions of the paper or possibly removed.

Motivated by this recommendation and other comments we have removed these results from the paper in order to make more room for the central claims of the manuscript (see Change 3 above).

Figure 4i seems to be the main demonstration that individual neural activity itself is not invariant to context. I'd like to see a more in–depth exploration of this. Particularly, if the readout only relied on a small handful of neurons then finding that the rest of the neurons are not context–invariant wouldn't prove that individual neural invariance is not a relevant mechanism. Given that the readout from this network is known, it would be particularly easy to determine if the heavily weighted neurons in particular are or are not context invariant.

We thank the reviewer for this suggestion on how to further explore the lack of invariance of single neurons. These analyses are now performed on the simpler network in a new Figure 4 (see Change 1 above). We have extended our analyses and the text according to the reviewer’s suggestions:

“However, a first inspection of neural activity indicated that single neurons are strongly modulated by context (Figure 4a). To quantify this, we determined the signal clarity for each neuron at each stage of the feedforward network, averaged across contexts (Figure 4b). As expected, the signal clarity was low for the sensory stimuli. Intriguingly, the same was true for all neurons of the modulated neural population, indicating no clean separation of the sources at the level of single neurons. Although most neurons had a high signal clarity in some of the contexts, there was no group of neurons that consistently represented one or the other source (Figure 4c). Furthermore, the average signal clarity of the neurons did not correlate with their contribution to the readout (Figure 4d).” (ll. 161-170)

In general, I don't understand why the authors use a separately trained linear readout when trying to show that the population activity at the final layer is invariant. They eventually acknowledge that "Since this readout is obtained from the data, this procedure does not require knowledge of the readout in the network model. Note that the trained decoder and the network readout are not necessarily identical" but they don't explain why they are using this alternative readout or what new insights its use adds. Particularly, the performance of the network indicates the there is some sort of context invariant read out possible from this population, yet the authors use this other readout in a way that is seemingly supposed to add something to the explanation.

We agree with the reviewer that it was not sufficiently clear why we used the linear readout obtained from the data. The original idea was to highlight that these analyses can be done on neural data, because they do not require knowing the readout that is performed by downstream areas. We acknowledge that raising this point while explaining the feedback-freezing-experiment is confusing. To address this, we now use the model’s readout (former Figure 6, now Figure 5). In addition, we explicitly highlight that an invariant readout could be obtained from neural data (see response to public review above).

Be sure to say what errorbars are based on in all figures.

Thank you for pointing this out. We have added the respective information to the captions.

"In our model, the mechanism needs to satisfy a few keyrequirements: i) the modulation is not uniform across the population, ii) it operates on a timescale similar to that of changes in context, and iii) it is driven by feedback projections." I don't understand claim (iii). If anything, the results show the importance of the modulation being driven by feedforward sensory signals (figure 2d/e).

Yes, that's a fair point. We have rephrased the respective sentence as follows:

“[…] iii) it is driven by a brain region that has access to the information needed to infer the context” (ll. 349-350)

"In addition, feedback inputs from the sensory to the modulatory system allow a better control of the modulated network state." I don't see how the connections from a sensory system to a modulatory system are "feedback".

We believe that our phrasing was unfortunate. We meant feedback from the feedforward network’s output. In the brain, this could correspond to higher-level sensory areas:

“In addition, feedback inputs from higher-level sensory areas to the modulatory system allow a better control of the modulated network state.” (ll. 375-377).

Reviewer #3:I appreciate the didactic way in which the manuscript was written (and beautiful figures!), in particular the progression from a vanilla architecture towards the full fledged model with EI rectified neurons with spatially specific modulation. My main concerns (detailed below) are two–fold:1. I felt that some extensions were not explicitly justified (e.g. why 2 layers instead of 2, etc)2. I was expecting more 'reverse–engineering' of the mechanism through which the network accomplishes a context invariant projection. This is the main result of the paper, as reflected in the title, so I think it deserves more unpacking. Below I unpack these concerns, sometimes providing some suggestions to improve the motivation and clarity of the paper (without any particular order)

We thank the reviewer for their very constructive feedback and agree with both points made here. We hope to have addressed them with the revision of the manuscript, in particular Changes 1-3 (see above). Below we provide answers to the specific recommendations and questions in some more detail.

1. Overall, the architecture choices are a bit unjustified. In the extreme, wouldn't the LSTM alone solve the task? The addition of each feedforward layer should be better motivated (e.g. more biologically realistic? In what sense?). For example, why add an extra layer from extensions 2 and 3? If those are necessary, this should be explained. If they are not necessary, they should be removed.

We are glad that the reviewers have brought this to our attention. We have substantially reorganised the paper to clarify which architectural choices are relevant for which finding (Changes 1 and 3).

We modified the description of the Dalean network to make our model choices more transparent to the reader:

“We extended the feedforward model as follows (Figure 6a): First, all neurons had positive firing rates. Second, we split the neural population (z in the previous model) into a "lower-level" (z^L^) and "higher-level" population (z^H^). […] This allowed for a direct comparison between a modulated and an unmodulated neural population. It also allowed us to include Dalean weights between the two populations.” (ll. 236-242)

“[…] the higher-level population contained a context-invariant subspace (Figure 6f). This was not the case for the lower-level population, underscoring that invariant representations do not just arise from projecting the sensory stimuli into a higher dimensional space.” (ll. 262-266)

Furthermore, we decided to perform our single neuron and population analyses on the simpler network with spatially diffuse modulation and verify them on the Dalean network (see Change 1). We hope that the new manuscript structure and the additional analyses address the concerns raised above.

Regarding the question whether an LSTM alone can solve the task: yes it can. We tested this during the course of the project. This is not surprising, because an LSTM could in principle even learn the same architecture as we use here. Our architecture may even make it more difficult to solve the task since it contains more constraints. However, the focus of this project was not on whether a recurrent network can solve this task, but rather what role feedback modulation could play in (invariant) sensory processing. We have therefore decided not to include any results from a purely recurrent network.

2. 'Because the task requires a dynamic inference of the context, it cannot be solved by feedforward networks or standard blind source separation algorithms' I think the paper could be better motivated if this was shown explicitly with some examples.

We agree that this would help to motivate the architecture of our model and have included a new supplementary figure, in which we explicitly demonstrate that a feedforward network cannot solve the task (see Supp. Figure S1).

3. A figure explicitly illustrating the training setup would help motivate what is trivially solved and what is actually challenging. For instance, in the main manuscript, it is not clear in which cases the network is trained and tested on the same contexts (ie A(t)) and which cases it is not. In the first case, the context can be easily inferred from x(t) but the latter is more challenging?

We thought this was an excellent idea and have created a methods figure that illustrates the task, the model and the training setup explicitly (see Figure 8). Generally, we always sample new contexts when testing the model, from the same distribution as during training (unless stated otherwise). We now also show that the network does not generalise to out-of-distribution contexts or sensory stimuli (Supp. Figure S6, see also Change 4 above).

4. however I understand that the paper is already too long, Intra / extrapolation results deserve more spotlight and unpacking in my opinion. In general, if there is a lack of space, I would merge Figure 1 and figure 2 – and jump directly to extension 1 – and move most of figure 2 to sup.

We thank the reviewer for this suggestion. We removed some results and reorganised the remaining results such that there is more focus on the invariant subspaces. Hence, we do not feel that the manuscript has become substantially longer. Furthermore, we would prefer to only show the proof of concept in the first figure, in order to not overload the reader. If the reviewers find that the paper is much too long, we could move figure 2 to the supplementary material.

5. Most important concern to me: Figure 6, in which the mechanism is revealed, deserves more quantifications to explicitly pinpoint the mechanism. Three suggestions come to mind:

Thanks, these are great suggestions.

a. Plot the 3 PCs components (instead of just 1) and show the readout in this space. The key result is that the readout is invariant to context and this is not clearly illustrated at the moment. Instead, what is shown is that the representation changes, but that it changes in a way that preserves invariance on the readout is not clearly highlighted.

Unfortunately, using the space of the first 3 PCs instead of the first PC and the readout axes does not illustrate the invariant subspace in an intuitive way (see Change 2). Therefore, we decided to stick with the more unconventional space but verify our findings for PC space in a new supplementary figure (Supp. Figure S9). To better illustrate that the readout is context-invariant we plotted the projection of the subspace onto the readout into the 3D figure (Figure 5b) and make explicit that this is the same projection as shown in Figure 5c.

b. The authors highlight that the network is not just reversing the new mixing coefficients and projecting the activity back into the 2d low manifold. Instead, it is rotating everything out of this manifold. My suggestion would be to show this alternatively explicitly. Is it actually possible? Relatedly, what happens if the context is changed back to context 1?c. Finally, all the statements made about this figure should be quantified and not just illustrated for 1 trial.

Regarding b and c: As described in Change 2, we have implemented the reviewer’s advice by adding a quantification to the former Figure 6 (now Figure 5) in terms of the angles between the subspaces (Figure 5d). We also show that the magnitude of this change depends on the similarity of the old and the new context (Figure 5dande), indicating that there is a consistent mapping between context and the low-dimensional population activity. Switching back to context 1 would therefore reinstate the original subspace, without hysteretic effects.